# LIMI: LESS IS MORE FOR AGENCY

## ABSTRACT

We define "*Agency*" as the emergent capacity of AI systems to function as autonomous agents—actively discovering problems, formulating hypotheses, and executing solutions through self-directed engagement with environments and tools. This fundamental capability marks the dawn of the "*Age of AI Agency*", driven by a critical industry shift: the urgent need for AI systems that **don't just think, but work**. While current AI excels at reasoning and generating responses, industries demand autonomous agents that can execute tasks, operate tools, and drive real-world outcomes. As agentic intelligence becomes the defining characteristic separating cognitive systems from productive workers, efficiently cultivating machine autonomy becomes paramount. Current approaches assume that more data yields better agency, following traditional scaling laws from language modeling. We fundamentally challenge this paradigm. **LIMI** (Less Is More for Intelligent Agency) demonstrates that agency follows radically different development principles. Through strategic focus on collaborative software development and scientific research workflows, we show that sophisticated agentic intelligence can emerge from minimal but strategically curated demonstrations of autonomous behavior. Using only 78 carefully designed training samples, LIMI achieves 73.5% on AgencyBench, dramatically outperforming state-of-the-art models: Kimi-K2-Instruct (24.1%), DeepSeek-V3.1 (11.9%), Qwen3-235B-A22B-Instruct (27.5%), and GLM-4.5 (45.1%). Most strikingly, LIMI demonstrates 53.7% improvement over models trained on 10,000 samples—achieving superior agentic intelligence with 128 times fewer samples.

Our findings establish the *Agency Efficiency Principle*: machine autonomy emerges not from data abundance but from strategic curation of high-quality agentic demonstrations. This discovery fundamentally reshapes how we develop autonomous AI systems, suggesting that **mastering agency requires understanding its essence, not scaling training data**. As industries transition from thinking AI to working AI, LIMI provides a paradigm for sustainable cultivation of truly agentic intelligence. Our data and code are available in an anonymous repository and will be made publicly available upon acceptance.

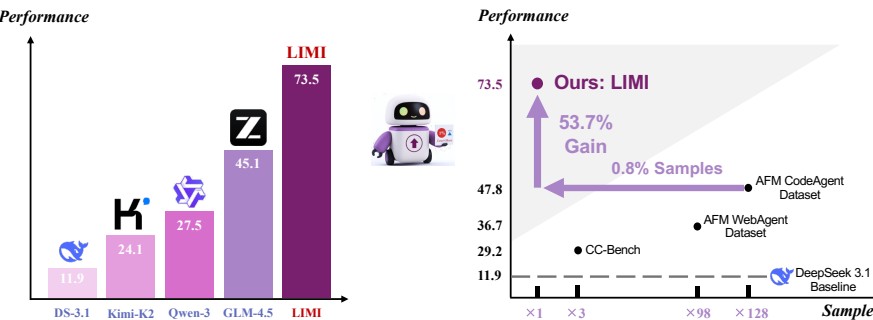

Figure 1: LIMI demonstrates the Less-Is-More principle for agentic intelligence. **Left:** LIMI achieves 73.5% performance on AgencyBench, outperforming all baseline models. **Right:** Using only 78 training samples, LIMI shows 53.7% improvement over models trained on 10,000 samples.

# 1 INTRODUCTION

The emergence of agentic Large Language Models (LLMs)–systems that can reason, act, and interact autonomously Locke (1987)–represents a paradigm shift from passive AI assistants to proactive intelligent agents (Wang et al., 2024; Xi et al., 2023). We define *Agency* as the emergent capacity of AI systems to function as autonomous agents–actively discovering problems, formulating hypotheses, and executing solutions through self-directed engagement with environments and tools. This fundamental capability marks the dawn of the *Age of AI Agency*, driven by a critical industry shift: the urgent need for AI systems that **don't just think, but work**. While current AI excels at reasoning and generating responses (Brown et al., 2020; Chowdhery et al., 2023; Touvron et al., 2023; OpenAI, 2023; Anil et al., 2023), industries demand autonomous agents that can execute tasks, operate tools, and drive real-world outcomes through capabilities like autonomous task execution (Qin et al., 2023; Yang et al., 2023; Parisi et al., 2022), multi-step reasoning (Wei et al., 2022; Yao et al., 2024b; Besta et al., 2024; Zhang et al., 2023b; Dhuliawala et al., 2023), and collaborative problem-solving (Li et al., 2023; Chan et al., 2023; Du et al., 2023; Zhang et al., 2023a; Qian et al., 2023).

However, the development of such agentic systems faces critical challenges. Current approaches assume that more data yields better agentic intelligence, following traditional scaling laws from language modeling (Kaplan et al., 2020; Rae et al., 2021; Chowdhery et al., 2023; Scao et al., 2022; Zhang et al., 2022). This paradigm leads to increasingly complex training pipelines and substantial resource requirements, yet this fundamental assumption remains largely untested: *do agentic capabilities truly require exposure to vast amounts of training data, or could they emerge more efficiently through strategic approaches?* Emerging evidence from adjacent domains suggests a compelling alternative paradigm. LIMA (Zhou et al., 2023) achieved effective model alignment with only 1,000 carefully curated examples, while LIMO (Ye et al., 2025) demonstrated that complex mathematical reasoning can emerge from just 817 strategically selected training samples, achieving a remarkable 45.8% absolute improvement with only 1% of the data typically required. These convergent findings suggest that strategic data curation may be fundamentally more powerful than dataset scale for developing sophisticated AI capabilities, naturally leading us to investigate whether agentic intelligence follows similar efficiency principles.

We introduce **LIMI** (**L**ess **I**s **M**ore for **I**ntelligent **A**gency), which demonstrates that agency follows radically different development principles from traditional scaling approaches. Through strategic focus on collaborative software development and scientific research workflows–domains that collectively span the majority of knowledge work scenarios–we show that sophisticated agentic intelligence can emerge from minimal but strategically curated demonstrations of autonomous behavior. Our approach is grounded in three core innovations: (1) First, we pioneer novel agentic user query synthesis methodologies, including human-AI collaborative query collection from real-world scenarios and systematic GitHub pull request-based query synthesis using advanced LLMs, ensuring that our training demonstrations capture authentic patterns of agentic behavior while maintaining ecological validity; (2) Second, we develop a systematic trajectory collection protocol that captures complete multi-turn interaction sequences for each curated query, recording the full collaborative workflow from initial task understanding through iterative model reasoning, tool utilization, and environmental feedback to successful task completion, providing high-quality training demonstrations of sophisticated agentic behavior in realistic operational contexts; (3) Third, we reveal the data efficiency principle for AI agency cultivation, demonstrating that sophisticated agentic intelligence emerges from strategic curation of minimal high-quality demonstrations rather than large-scale data accumulation, fundamentally challenging traditional scaling paradigms in agentic AI development.

Using only 78 carefully designed training samples, LIMI achieves 73.5% on comprehensive agency benchmarks, dramatically outperforming state-of-the-art models: Kimi-K2-Instruct (24.1%), DeepSeek-V3.1 (11.9%), Qwen3-235B-A22B-Instruct (27.5%), and GLM-4.5 (45.1%). Most strikingly, LIMI demonstrates 53.7% improvement over models trained on 10,000 samples–achieving superior agentic intelligence with 128 times fewer samples. These findings establish the *Agency Efficiency Principle*: machine autonomy emerges not from data abundance but from strategic curation of high-quality agentic demonstrations. This discovery fundamentally reshapes how we develop autonomous AI systems, suggesting that mastering agency requires understanding its essence, not scaling training data. As industries transition from thinking AI to working AI, LIMI provides a paradigm for sustainable cultivation of truly agentic intelligence, demonstrating that the key to effective agentic AI development lies in strategic data curation rather than computational scale.

## 2 PRELIMINARY

The emergence of agentic AI systems marks a fundamental shift from passive assistants to autonomous intelligent agents. We define *Agency* as the capacity to autonomously discover problems, formulate hypotheses, and execute solutions through self-directed engagement with environments and tools. This evolution demands AI systems that **don't just think, but work**, which requires integration of autonomous execution, multi-step reasoning, and collaborative problem-solving.

### 2.1 LONG-HORIZON TASKS AND AGENTIC COMPLEXITY

Agentic intelligence is fundamentally tested through complex, multi-step challenges requiring sustained cognitive effort across extended interaction sequences. As illustrated in Figure 5, these scenarios demand sophisticated capability integration spanning autonomous task execution, multi-step reasoning, and collaborative problem-solving across diverse domains like software development and scientific research. Such tasks exhibit several key characteristics: **temporal complexity** through multi-round interactions requiring coherent state tracking and cumulative reasoning; **strategic planning** that decomposes complex objectives into manageable sub-goals while adapting based on environmental feedback; **tool orchestration** requiring coordinated use of multiple systems with integrated result processing; and **collaborative communication** ensuring effective human-AI coordination throughout extended problem-solving processes. These characteristics distinguish genuine agentic intelligence from passive AI systems that merely respond to individual queries.

### 2.2 DOMAIN SPECIFICATION: VIBE CODING AND RESEARCH WORKFLOWS

We focus on two fundamental domains that collectively span the majority of knowledge work scenarios and require the full spectrum of agentic capabilities. (1) **Vibe Coding** represents collaborative software development where AI agents work alongside human in natural, context-rich environments. This domain demands: code understanding and generation across existing codebases; navigation through complex development tool ecosystems; iterative problem-solving through debugging and optimization cycles; and collaborative communication for technical coordination. The complexity lies in holistic understanding of development contexts and principled decision-making under evolving requirements. (2) **Research Workflows** encompass scenarios where agents navigate complex scientific processes, including literature search, data analysis, experiment design, and insight generation. These workflows require: information synthesis from diverse sources; experimental design with appropriate methodologies; data analysis and interpretation of complex results; and knowledge communication across different stakeholder formats. Research workflows demand sophisticated reasoning capabilities spanning from creative hypothesis generation to rigorous analytical execution.

## 3 DATASET CONSTRUCTION

The effectiveness of the LIMI approach relies fundamentally on strategic data curation that captures essential agentic behaviors through real-world collaborative tasks. As illustrated in Figure 2, this section describes our comprehensive methodology for constructing training data that validates the Less-Is-More hypothesis for agentic intelligence, focusing on the systematic collection and curation of real-world collaborative tasks.

### 3.1 FRAMEWORK AND NOTATION

We formalize our data construction process around the fundamental elements of agentic interaction, defining each complete interaction as a tuple $(q_i, \tau_i)$ where queries initiate collaborative workflows and trajectories capture complete interaction sequences.

**Query Definition** We begin with the *query* $q_i$ as the foundational element that initiates agentic interaction. Each query represents a natural language specification from the user that articulates the desired objective, ranging from software development requirements in vibe coding scenarios to research tasks in scientific workflows. The query establishes both the starting point and success criteria for the subsequent collaborative process.

Figure 2: LIMI Data Construction Pipeline. **Left**: user query pool construction through GitHub PR synthesis and real-world query collection with quality review. **Right**: Trajectory collection via human-AI collaboration in SII CLI environment, capturing complete interaction sequences.

**Trajectory Formalization** The *trajectory* $\tau_i = \{a_{i,1}, \ldots, a_{i,n_i}\}$ captures the subsequent collaborative trajectory following the initial query. Each action $a_{i,j}$ in the trajectory represents one of three fundamental interaction types that constitute the agentic response process. **Model reasoning** ($m_{i,j}$) captures the agentic model's reasoning output, demonstrating understanding, analysis, planning, and decision-making processes. **Model tool calling** ($t_{i,j}$) represents structured tool invocations executed by the model to interact with external environments. **Environment observation** ($o_{i,j}$) includes results and outputs returned from tool executions, as well as user feedback and clarifications provided during the collaborative process. The sequential index $j$ maintains the temporal ordering of these interactions within trajectory $i$, and $n_i$ denotes the total number of actions.

## 3.2 QUERY POOL CONSTRUCTION

Our query collection strategy combines authentic real-world scenarios with systematically expanded coverage to ensure both ecological validity and sufficient training diversity for agentic intelligence development.

**Real-world Query Collection** We collect 60 queries from actual scenarios encountered by professional developers and researchers in collaborative environments. These queries represent authentic challenges that arise in real-world software development and research workflows, capturing the natural complexity and contextual richness of human-AI collaborative work across both domains. Notably, a substantial portion of the research queries are derived from real academic papers (Xiao et al., 2025a;b; Jiang et al., 2025; Li et al., 2025a; Sun et al., 2025; 2024), ensuring authentic representation of genuine research challenges.

**GitHub PR-based Query Synthesis** To systematically expand our query pool while preserving authenticity, we develop a pipeline for synthesizing queries from GitHub Pull Requests (PRs) using GPT-5 (OpenAI). This approach leverages real code changes to generate authentic collaborative scenarios that reflect genuine development needs. The prompt can be found in Appendix C.

Our systematic curation process involves five key stages: (1) **Repository Selection**: We select 100 repositories with over 10,000 GitHub stars to ensure high-quality codebases representing industry best practices. (2) **Domain Diversification**: We ensure comprehensive coverage across diverse software development domains. (3) **Complexity Filtering**: We filter PRs by unified diff patch token count (below 1,200 tokens) and exclude Markdown-only modifications, focusing on substantive code changes requiring meaningful agentic intervention. (4) **Scale and Sampling**: From each repository, we collect 1,000 PRs and randomly sample 100 for query synthesis, ensuring statistical representativeness. (5) **Quality Assurance**: We randomly sample 200 queries from the total of 10,000 synthesized queries and employ four PhD students in computer science as expert annotators for quality evaluation. The evaluation criteria focus on semantic alignment between the generated query and the corresponding PR content, ensuring that synthetic queries accurately capture the intent and context of real development scenarios. Additionally, we require that the generated queries correspond to our designated domains, as illustrated in Figure 3. If any annotator determines that a query fails to meet these requirements, the query is discarded. Through this rigorous filtering process, 18 high-quality queries are retained for our final dataset.

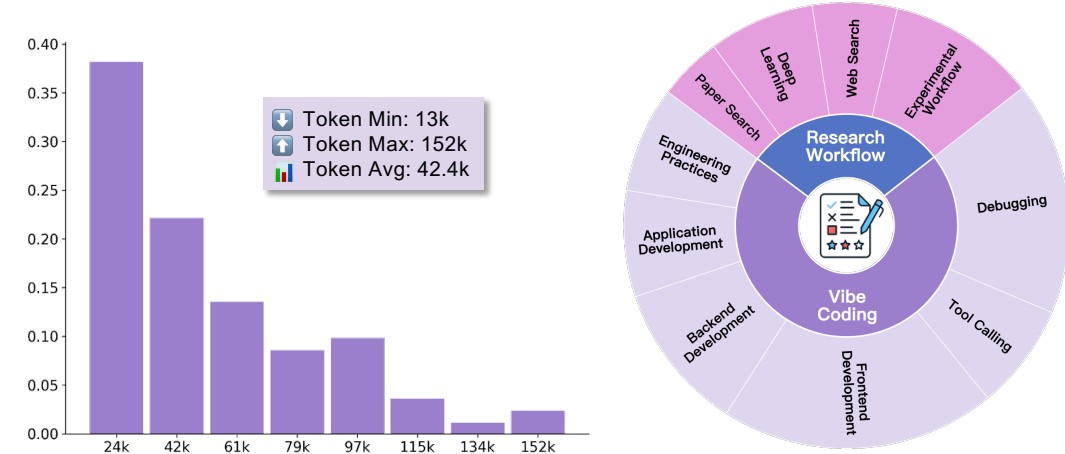

Figure 3: Characteristics of LIMI's training data. **Left**: Trajectory length distribution showing interaction complexity (average 42.4k tokens). **Right**: Domain coverage across vibe coding and research workflows.

Through this systematic approach, we assemble a comprehensive query pool $\mathcal{Q} = \{q_1, q_2, \ldots, q_{78}\}$ consisting of 78 high-quality queries. This strategically curated query pool forms the foundation of LIMI training data. Each query reflects authentic collaborative scenarios, ensuring that the dataset captures the full spectrum of challenges encountered in professional development and research.

### 3.3 TRAJECTORY COLLECTION FOR TRAINING DATASET

To generate training trajectories that demonstrate optimal agentic behavior and validate the real-world effectiveness of LIMI approach, we require a sophisticated execution environment that enables authentic human-AI collaboration. This environment must support complex tool interactions, maintain detailed interaction logging, and provide the operational context necessary for realistic agentic intelligence assessment.

**Execution Environment Selection** Several command line interface (CLI) environments are available for agentic models, including Claude Code[1], Gemini CLI[2], and SII CLI (Lin et al., 2025). We select SII CLI based on four critical advantages: (1) comprehensive tool integration supporting both vibe coding and research workflows, (2) detailed trajectory logging for high-quality training data collection, (3) flexible human-AI collaboration interfaces enabling natural interaction patterns. SII CLI ensures our collected trajectories capture realistic collaborative contexts, making it well-suited for validating our hypothesis that strategic data curation yields superior agentic intelligence.

**Controlled Trajectory Collection Protocol** We employ a systematic data collection protocol designed to capture optimal agentic behavior patterns. Four PhD student annotators serve as human collaborators, working alongside GPT-5 as the agentic model to complete the 78 queries in realistic collaborative scenarios. For each query $q_i$, we employ an iterative collection approach, continuously gathering trajectories until successful completion is achieved. This persistent methodology ensures that collected trajectories capture authentic human-AI interaction patterns, including natural back-and-forth communication, iterative refinement processes, and collaborative problem-solving strategies that characterize effective agentic behavior.

The data collection process captures complete interaction sequences for each query $q_i$, generating the corresponding trajectory $\tau_i = \{a_{i,1}, \ldots, a_{i,n_i}\}$ that includes all fundamental interaction types: model reasoning ($m_{i,j}$), model tool calling ($t_{i,j}$), and environment observation ($o_{i,j}$). As shown in Figure 3, these trajectories capture extensive interaction sequences, with the longest trajectory reaching 152k tokens, demonstrating the depth and complexity of collaborative problem-solving

---

[1]https://github.com/anthropics/claude-code
[2]https://github.com/google-gemini/gemini-cli

| Model | Model Size | Samples | Agency Bench FTFC | Agency Bench RC@3 | Agency Bench SR@3 | AVG. |
|---|---|---|---|---|---|---|
| Qwen3-4B | 4B | N/A | 3.7 | 3.7 | 6.5 | 4.6 |
| Qwen3-8B | 8B | N/A | 3.7 | 5.1 | 13.0 | 7.3 |
| Qwen3-32B | 32B | N/A | 6.5 | 9.3 | 9.3 | 8.4 |
| DeepSeek-V3.1 | 671B | N/A | 10.6 | 11.9 | 13.3 | 11.9 |
| Kimi-K2-Instruct | 1T | N/A | 20.7 | 25.1 | 26.6 | 24.1 |
| Qwen3-235B-A22B-Instruct | 235B | N/A | 23.0 | 28.2 | 31.3 | 27.5 |
| GLM-4.5 | 355B | N/A | 37.8 | 50.0 | 47.4 | 45.1 |
| GPT-5 | N/A | N/A | 56.1 | 59.4 | 62.8 | 59.4 |
| LIMI | 355B | 78 | **71.7** | **74.2** | **74.6** | **73.5** |
| *Data Efficiency* | | | | | | |
| GLM-4.5-CC | 355B | 260 | 30.4 | 30.4 | 26.7 | 29.2 |
| GLM-4.5-Web | 355B | 7,610 | 36.7 | 36.7 | 36.7 | 36.7 |
| GLM-4.5-Code | 355B | 10,000 | 48.0 | 48.0 | 47.5 | 47.8 |
| LIMI | 355B | 78 | **71.7** | **74.2** | **74.6** | **73.5** |

Table 1: Comprehensive comparison of models on AgencyBench. Models are grouped by evaluation purpose: baseline comparisons, generalization assessment, and data efficiency validation.

processes that characterize sophisticated agentic behavior. This approach guarantees that our models learn not only from successful outcomes but also from the complete problem-solving process, including how to adapt strategies and recover from failures during collaborative execution.

### 3.4 EVALUATION BENCHMARKS

Our evaluation framework encompasses two complementary assessments to comprehensively validate the effectiveness of the LIMI approach across diverse agentic scenarios. (1) **AgencyBench** We evaluate all models on AgencyBench (Li et al., 2025b), a comprehensive evaluation benchmark specifically designed for assessing agentic capabilities in collaborative scenarios. The tasks in AgencyBench are illustrated in Table 3. AgencyBench contains carefully curated tasks that reflect the complexity and collaborative nature of real-world agentic scenarios across both vibe coding and research workflows, providing a rigorous test of the LIMI hypothesis that strategic data curation yields superior agentic intelligence. This benchmark enables direct measurement of models' performance on representative queries that mirror the authentic collaborative contexts. (2) **Out-of-domain Benchmarks** To assess generalization capabilities beyond our core domains, we evaluate model performance on established benchmarks spanning diverse agentic and coding scenarios: TAU2-bench-airline and TAU2-bench-retail (Yao et al., 2024a; Barres et al., 2025) for tool use capabilities, EvalPlus-HumanEval and EvalPlus-MBPP (Liu et al., 2024; 2023) for code generation performance, DS-1000 (Lai et al., 2022) for data science and code generation tasks, and SciCode (Tian et al., 2024) for scientific computing applications. This comprehensive evaluation suite ensures that our findings generalize beyond the specific domains of vibe coding and research workflows.

## 4 EXPERIMENTS

### 4.1 EXPERIMENT SETUP

**Baseline Models**  We evaluate against a diverse set of state-of-the-art foundation models to ensure comprehensive comparison: GLM-4.5 (Zeng et al., 2025), GLM-4.5-Air (Zeng et al., 2025), Qwen3-4B, Qwen3-8B, Qwen3-32B, Qwen3-235B-A22B-Instruct(Yang et al., 2025), DeepSeek-V3.1 (Yang et al., 2025), Kimi-K2-Instruct (Team et al., 2025). This selection encompasses open-source models with varying architectural designs and training methodologies, enabling a rigorous evaluation of agentic capabilities.

**Model Training and Variants**  To systematically evaluate the impact of our curated training data, we fine-tune Qwen3-4B, Qwen3-8B, Qwen3-32B, GLM-4.5, and GLM-4.5-Air using our training dataset. Additionally, to assess the quality and effectiveness of our data curation strategy, we conduct comparative experiments by fine-tuning GLM-4.5 on three alternative datasets: CC-Bench-trajectories (Zeng et al., 2025), AFM-WebAgent-SFT-Dataset (PersonalAILab, 2024), and AFM-

| Model | TAU2-bench-airline | TAU2-bench-retail | DS-1000 | EvalPlus-HE | EvalPlus-MBPP | SciCode-MP | SciCode-SP | AVG. |
|---|---|---|---|---|---|---|---|---|
| Qwen3-4B | 8.0 | 5.0 | 16.7 | 85.4 | 72.5 | 0.0 | 10.4 | 28.3 |
| Qwen3-8B | 12.0 | 7.0 | 22.3 | 86.0 | 73.5 | 0.0 | 17.7 | 31.2 |
| DeepSeek-V3.1 | 32.0 | 6.1 | **42.4** | 82.3 | 68.3 | 0.0 | 7.3 | 34.1 |
| Qwen3-32B | 12.0 | 10.5 | 28.9 | 91.5 | 76.5 | 3.1 | 24.0 | 35.2 |
| Qwen3-235B-A22B-Instruct | 20.0 | 16.7 | 39.3 | 90.2 | 81.7 | 0.0 | 22.6 | 38.6 |
| Kimi-K2-Instruct | **38.0** | 28.9 | 23.1 | 92.1 | 77.5 | 3.1 | 23.6 | 40.9 |
| GLM-4.5 | 28.0 | 36.8 | 33.6 | 90.2 | 79.6 | 1.5 | 25.3 | 42.1 |
| GPT-5 | 26.0 | 18.4 | 40.8 | 91.5 | 81.7 | **10.8** | **33.3** | 43.2 |
| LIMI | 34.0 | **45.6** | 36.6 | 92.1 | 82.3 | 3.1 | 25.3 | **45.6** |
| *Data Efficiency* | | | | | | | | |
| GLM-4.5-Web | 18.0 | 13.2 | 33.9 | 84.1 | 75.1 | 0.0 | 2.8 | 32.4 |
| GLM-4.5-CC | **38.0** | 39.6 | **38.7** | 90.2 | 80.2 | 3.1 | 14.6 | 43.5 |
| GLM-4.5-Code | 20.0 | 16.7 | 38.5 | 87.8 | 78.3 | 3.1 | 21.5 | 38.0 |
| LIMI | 34.0 | **45.6** | 36.6 | **92.1** | **82.3** | **3.1** | **25.3** | **45.6** |

Table 2: Comprehensive performance comparison across out-of-domain benchmarks. HE represents EvalPlus-HumanEval, while MP and SP represent Main Problem and Sub Problem metrics of Sci-Code, respectively.

CodeAgent-SFT-Dataset (PersonalAILab, 2024). This experimental design enables direct comparison between our strategically curated data and existing large-scale agentic training datasets, providing empirical evidence for the Less-Is-More hypothesis. All fine-tuning experiments are conducted using the slime framework [3] with identical training configurations to ensure fair comparison.

For clarity, we refer to models fine-tuned with our curated dataset as LIMI (corresponding to fine-tuning GLM-4.5) and LIMI-Air (corresponding to fine-tuning GLM-4.5-Air). Models trained on alternative datasets are denoted with corresponding suffixes: CC for CC-Bench-trajectories dataset, Web for AFM-WebAgent-SFT-Dataset, and Code for AFM-CodeAgent-SFT-Dataset.

## 4.2 MAIN RESULTS

**Superior Performance on AgencyBench Against State-of-the-Art Models** Our experimental results demonstrate that LIMI achieves substantial performance advantages over leading foundation models on agentic intelligence tasks. As shown in Table 1, LIMI achieves an impressive average score of 73.5% on AgencyBench, significantly outperforming all baseline models: GPT-5 (59.4%), GLM-4.5 (45.1%), Qwen3-235B-A22B-Instruct (27.5%), Kimi-K2-Instruct (24.1%), and DeepSeek-V3.1 (11.9%). The performance gap is particularly pronounced in first-turn functional completeness (FTFC), where LIMI achieves 71.7% compared to the best baseline performance of 56.1% from GPT-5, representing a remarkable 15.6 percentage point improvement. Similarly, LIMI demonstrates superior task completion reliability with 74.6% success rate, substantially exceeding the 62.8% achieved by GPT-5, the strongest baseline model.

**Consistent Advantages Across Diverse Out-of-domain Benchmarks** LIMI's superiority extends across established benchmarks spanning tool use, coding, and scientific computing domains (Table 2). With an average performance of 45.6%, LIMI outperforms all baseline models. Notably, LIMI achieves the highest performance on critical coding benchmarks (EvalPlus-HumanEval: 92.1%, EvalPlus-MBPP: 82.3%) and demonstrates competitive results on tool use tasks (TAU2-bench-airline: 34.0%, TAU2-bench-retail: 45.6%). The consistent performance advantages demonstrate that our strategic data curation approach yields broad improvements in model capabilities, establishing strong performance across multiple domains beyond our core vibe coding and research workflows focus.

## 4.3 DATA EFFICIENCY ANALYSIS

**Strategic Curation Dramatically Outperforms Scale-Based Approaches** Our experimental results provide compelling empirical evidence for the core LIMI hypothesis that strategic data curation is fundamentally more effective than simply scaling training data volume for developing agentic intelligence. As demonstrated in Tables 1 and 2, LIMI achieves exceptional performance using only

---

[3] https://github.com/THUDM/slime

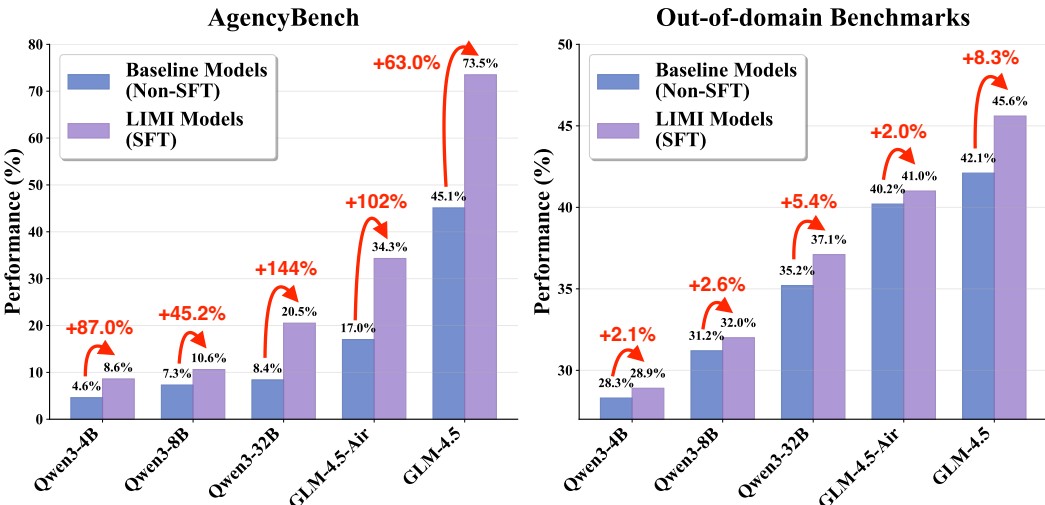

Figure 4: Performance comparison before and after LIMI fine-tuning across different models. LIMI-trained models demonstrate consistent improvements on both AgencyBench (left) and out-of-domain benchmarks (right).

78 carefully curated training samples, substantially outperforming models trained on datasets that are orders of magnitude larger. Most striking is the comparison with GLM-4.5-Code: LIMI's 73.5% average AgencyBench performance dramatically exceeds the 47.8% achieved by the large-scale approach (25.7 percentage point improvement), despite using a dataset 128 times smaller. Similar advantages extend to out-of-domain benchmarks, where LIMI (45.6%) outperforms GLM-4.5-Code (38.0%), GLM-4.5-Web (32.4%), and GLM-4.5-CC (43.5%) with significantly fewer samples. These consistent improvements validate that strategic data curation enables more effective capability transfer than large-scale data collection, establishing the Less-Is-More paradigm as broadly applicable to agentic intelligence development.

## 4.4 GENERALIZATION ANALYSIS

Our approach demonstrates remarkable effectiveness across multiple dimensions of generalization, validating the broad applicability of strategic data curation for agentic intelligence development.

**Cross-Scale Generalization** As shown in Figure 4, LIMI consistently improves performance across different model sizes within the same family. The Qwen3 series demonstrates substantial relative improvements from +87.0% (4B) to +45.2% (8B) to +144% (32B) on AgencyBench. Similarly, the GLM family shows robust improvements with GLM-4.5-Air (106B parameters) achieving +102% relative improvement (17.0% → 34.3%) and GLM-4.5 (355B parameters) reaching +63.0% improvement (45.1% → 73.5%). Even the smallest models benefit substantially, with Qwen3-4B nearly doubling its performance (4.6% → 8.6%), demonstrating effectiveness across the entire spectrum of model scales.

**Cross-Architecture Generalization** The effectiveness extends across fundamentally different architectural paradigms, demonstrating that LIMI's strategic data curation methodology is architecture-agnostic. Dense transformer models (Qwen3 series) and mixture-of-experts architectures (GLM series) both exhibit substantial improvements. This cross-architectural consistency indicates that our strategic data curation captures fundamental agentic behavioral patterns that are independent of specific model implementations, parameter distribution strategies, or computational architectures, establishing broad compatibility across diverse foundation model designs.

**Cross-Domain Generalization** Beyond our target domains of vibe coding and research workflows, LIMI demonstrates consistent improvements on out-of-domain benchmarks spanning tool use, coding, and scientific computing tasks. The relative improvements range from +2.1% (Qwen3-

4B) to +8.3% (GLM-4.5), with all models showing positive gains despite the evaluation tasks being outside our training domain focus. This phenomenon suggests that our strategic data curation enhances fundamental reasoning and problem-solving capabilities rather than merely optimizing for specific task types.

## 5 RELATED WORK

### 5.1 AGENTIC LANGUAGE MODEL

The development of agentic language models signifies a fundamental paradigm shift from passive text generation to autonomous decision-making systems. Early key contributions laid the foundation for modern agentic capabilities: Schick et al. (2023) introduced Toolformer, demonstrating that language models can learn to use external tools via APIs in a self-supervised manner, while Yao et al. (2023) proposed ReAct, which synergizes reasoning and acting by enabling LLMs to generate both reasoning traces and task-specific actions in an interleaved manner. The emergence of autonomous systems like Significant Gravitas (2025) marked the transition toward fully autonomous agents capable of breaking down high-level goals into manageable subtasks and executing them independently.

Building upon these early breakthroughs, the latest research has shifted its focus to foundational models specifically designed for agentic capabilities. GLM-4.5 (Zeng et al., 2025) provides a unified approach to reasoning, coding, and agentic tasks, featuring hybrid reasoning modes and achieving 90.6% tool-calling success rate. Similarly, Kimi-K2 (Team et al., 2025) introduces a trillion-parameter MoE architecture specifically optimized for agentic intelligence with native tool use capabilities and verifiable reward training. The integration of reinforcement learning has become central to developing robust agentic systems, with Zhang et al. (2025) formalizing this evolution as the transition from single-step preference optimization to temporally extended, partially observable Markov decision processes. However, current training methodologies—including those employed by state-of-the-art models like GLM-4.5 and Kimi-K2—predominantly rely on large-scale data synthesis and extensive computational resources. Our work addresses this limitation by demonstrating that sophisticated agentic capabilities can emerge from strategically curated minimal data, offering a more efficient path toward developing capable agentic systems.

### 5.2 DATA EFFICIENCY IN LANGUAGE MODELS

The paradigm of data efficiency in language models has gained significant attention as researchers recognize that strategic data curation can yield superior results compared to scaling training data volume. Zhou et al. (2023) demonstrate that with just 1,000 carefully curated prompts and responses, models can achieve effective alignment and generalize to diverse tasks. Building upon this foundation, Ye et al. (2025) extend this paradigm to complex mathematical reasoning, achieving a remarkable 45.8% improvement with only 817 strategically selected training samples. However, applying these data efficiency principles to agentic intelligence—where systems must autonomously discover problems, formulate hypotheses, and execute solutions through collaborative engagement with environments and tools—remains unexplored. Our work bridges this gap by extending the Less-Is-More paradigm to autonomous agents, demonstrating that sophisticated agentic capabilities can emerge from strategically curated demonstrations of collaborative behavior.

## 6 CONCLUSION

We demonstrate that sophisticated agentic capabilities can emerge through minimal but strategically curated demonstrations of autonomous behavior rather than massive dataset scaling. Using only 78 carefully designed training samples focused on vibe coding and research workflows, our approach achieves 73.5% performance on AgencyBench, substantially outperforming models trained on datasets up to 128 times larger. These findings establish the Agency Efficiency Principle: machine autonomy emerges not from data abundance but from strategic curation of high-quality agentic demonstrations. Our work fundamentally challenges conventional scaling paradigms in agentic AI development, demonstrating that mastering agency requires understanding its essence rather than accumulating training data.

REPRODUCIBILITY STATEMENT

We are committed to ensuring the reproducibility of our LIMI research. To facilitate replication and validation of our results, we provide comprehensive resources and detailed documentation across multiple dimensions.

**Code and Implementation:** Complete data processing scripts, model training code, and all experimental configurations are available in the anonymous repository at `https://anonymous.4open.science/r/limi-2F47`. This repository includes the GitHub PR-based query synthesis pipeline, trajectory collection protocols, fine-tuning scripts using the slime framework, and evaluation code for all benchmarks. Model parameters and hyperparameters used in the experiments are fully documented and provided.

**Dataset and Evaluation:** Our strategically curated 78-sample training dataset, including both real-world queries and GitHub PR-synthesized queries, is included in the repository at `https://anonymous.4open.science/r/limi-2F47`. The AgencyBench evaluation tasks and metrics are comprehensively documented in Appendix B and available through the official benchmark website at `https://agencybench.opensii.ai/`. For all generalization benchmarks (TAU2-bench, evalplus, DS-1000, SciCode), we strictly adhere to the original benchmark configurations without modifications, ensuring direct comparability with existing literature.

**Experimental Setup:** Section 4.1 provides complete experimental configurations, including baseline model specifications, training procedures, and evaluation protocols. The SII CLI environment setup and human-AI collaboration protocols for trajectory collection are detailed in Section 3.3.

These resources collectively provide the foundation for independent verification and extension of the LIMI approach, supporting the broader research community in advancing agentic intelligence through strategic data curation.

ETHICS STATEMENT

This research adheres to the ICLR Code of Ethics and addresses potential ethical considerations in our methodology and data collection processes.

**Human Subjects:** Our trajectory collection involves PhD students as voluntary annotators who collaborate with AI systems. All participants provide informed consent, are fully aware of the research purpose, and can withdraw at any time. The process does not involve sensitive personal information collection or potential harm to participants.

**Data Privacy and GitHub Usage:** Our GitHub PR-based query synthesis uses exclusively publicly available open-source repositories with over 10,000 stars. We do not collect private repository information, personal identifiers, or sensitive code. All synthetic queries are sufficiently abstracted to avoid reproducing specific implementation details while capturing general development patterns. Our GitHub data usage complies with platform terms of service and respects open-source community principles.

**Responsible Development:** We recognize the importance of responsible development of agentic AI capabilities. Our methodology focuses on improving collaborative and problem-solving abilities rather than creating potentially harmful autonomous behaviors.

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

## A  AGENTIC TASK COMPLEXITY ILLUSTRATION

Figure 5 demonstrates the inherent complexity of the agentic tasks addressed in our study. This Gomoku Battle application development query exemplifies how a single user request encompasses multiple interconnected subtasks that span diverse technical domains and cognitive requirements. The task demands sophisticated integration of web frontend development (UI rendering and interaction logic), advanced algorithmic implementation (game state management and AI search algorithms), and collaborative problem-solving capabilities (human-AI interaction design and multi-level difficulty scaling).

Each subtask requires distinct expertise: Subtask 1 involves frontend engineering with precise UI constraints; Subtask 2 demands algorithmic thinking for game logic and win detection; Subtask 3 requires state management and user experience design; Subtask 4 involves AI algorithm implementation with performance constraints; and Subtask 5 necessitates advanced search algorithms and optimization techniques. The complexity extends beyond individual components to their seamless integration within a unified application framework.

This example illustrates why strategic curation of high-quality demonstrations proves more effective than large-scale data accumulation for agentic intelligence development. Each carefully designed trajectory capturing the complete resolution of such multifaceted challenges provides dense concentrations of the strategic planning, technical execution, and collaborative refinement capabilities that define genuine agentic intelligence.

## B  AGENCY BENCH

### B.1  TASK OVERVIEW

To comprehensively evaluate agentic capabilities across diverse domains, we utilize AgencyBench (Li et al., 2025b), a carefully designed benchmark that encompasses both vibe coding and research workflow scenarios. AgencyBench consists of 10 distinct tasks that progressively test different aspects of agentic intelligence, from basic software development to complex scientific analysis. These tasks are specifically designed to require sustained multi-step reasoning, strategic tool utilization, and collaborative problem-solving—the core capabilities that define genuine agency.

The benchmark tasks can be categorized into two primary domains that reflect real-world knowledge work scenarios. The vibe coding domain includes collaborative software development tasks (Tasks 1-4) that require code generation, debugging, system architecture, and iterative development processes. The research workflow domain encompasses scientific investigation tasks (Tasks 5-10)

**User Query Example: Gomoku Battle - From Basics to Expert AI** ⚅

**Subtask 1: Web Frontend Development (UI/Rendering)**
*Goal:* Implement a 15×15 board rendering, black and white alternate moves (no win detection yet).
*Constraints:* Only native HTML/CSS/JS; no third-party libraries. Board supports click-to-place, disallowing stones on occupied points. Provide a "Reset" button.
*Deliverables:* Files: index.html, styles.css, app.js. Board as equal-spaced grid (Canvas or DOM).
*Acceptance Criteria:* Black moves first. Each valid click places a stone on the nearest intersection. No duplicate moves on the same intersection. After reset, board clears and black starts again.

**Subtask 2: Advanced Data Filtering & Sorting**
*Goal:* On top of Task 1, add detection of five in a row (horizontal, vertical, both diagonals). Highlight winning sequence and lock board.
*Constraints:* After victory, forbid further moves; "Reset" starts a new game. Detection algorithm should be O(1) incremental neighborhood check or O(n) range (no full scan).
*Deliverables:* Highlight 5 winning stones with line or glow. Display "Black / White Wins" at page top. Detection code in standalone function: checkWin(lastMove).
*Acceptance Criteria:* Immediate win when 5 in a row is formed. Six or more in a row still counts as win (standard Gomoku rule). Edge-of-board wins are detected correctly. Clicking occupied or locked board is invalid.

**Subtask 3: State Management (Undo/Redo/Replay)**
*Goal:* Support local two-player game management: move history, undo/redo, and step-by-step replay.
*Constraints:* Maintain moves[] stack, each element includes coordinates and color. Undo allows branching (history truncates). After game ends, replay and restart still available.
*Deliverables:* Buttons: Undo, Redo, Replay (more than 300ms per move). Board edges show coordinates (A–O / 1–15).
*Acceptance Criteria:* Undo back to opening without error. Redo returns to latest step. During replay, no manual moves allowed. After replay ends, normal play resumes. Undoing past winning move unlocks board.

**Subtask 4: Simple Rule-Based/Heuristic AI**
*Goal:* Add Human vs AI mode with simple AI. Easy: Random legal moves, prefer central 7×7. Medium: If winning in 1 move → take it. Else block opponent's open four. Else use scoring (open three > blocked three > open two).
*Constraints:* Mode selection: Local PvP / Human vs AI (choose who moves first). AI must decide within 100ms on empty 15×15 board.
*Deliverables:* Dropdown for mode and first player. Status bar: "AI Thinking...". AI function: aiMove(level); scoring function modularized.
*Acceptance Criteria:* Medium AI blocks human's "open four". Medium AI takes immediate winning move. Easy AI significantly weaker (Medium more than 70 percent win rate over Easy).

**Subtask 5: Advanced AI (Search Algorithms, e.g., Minimax)**
*Goal:* Implement stronger AI difficulty with performance control. Hard: Minimax + Alpha-Beta, fixed depth (2–3 ply), candidate pruning (recent moves, top K scoring). Expert: Based on Hard, add iterative deepening, time slicing (e.g. 500ms cutoff), transposition table (hash caching), killer move heuristic.
*Constraints:* Provide unified time/node count metrics in UI (e.g. "depth d=3, nodes n=12,345, time=0.43s"). Search must obey time limit; return best evaluation so far.
*Deliverables:* Difficulty: Easy / Medium / Hard / Expert; selectable first player. Debug panel (collapsible): eval score, candidate list (top K), search stats. Clear function layers: evaluate(board, player), generateCandidates(board), search(root, timeLimit).
*Acceptance Criteria:* Hard/Expert prioritize defense against "open four" and "double threats". Expert expands more nodes than Hard within 500ms and achieves higher win rate. On typical attack/defense test cases, Expert matches or approximates reference solutions.

Figure 5: An example of the user query, illustrating how a single query encompasses multiple interconnected subtasks across planning, execution, and collaboration dimensions, demonstrating the density of learning signals in high-quality demonstrations.

that demand literature analysis, data processing, experimental design, and insight synthesis. Each task contains multiple progressive subtasks of increasing complexity, ensuring comprehensive assessment of agentic capabilities from initial problem understanding through successful completion.

Table 3 presents a overview of all AgencyBench tasks, illustrating the breadth and depth of challenges that modern agentic systems must navigate. These tasks collectively span the spectrum of knowledge work scenarios, from building complex software systems with advanced features to conducting sophisticated research analyses with multiple data sources and evaluation metrics. The diversity and authenticity of these tasks make AgencyBench an ideal testbed for validating the LIMI hypothesis that strategic data curation can effectively cultivate agentic intelligence.

Below, we provide the details of all 10 examples from AgencyBench, including 4 vibe coding and 6 vibe research tasks. Each task contains several subtasks. Among them, tasks 6 and 8 include some raw code and data files, but we only present the queries.

## B.2 VIBE CODING

---

### Task 1: C++ Console Chat System

**Project: Advanced C++ Console Chat System**

**General Rules:**
- All code should be placed under `workspace/`.
- Use `g++ -std=c++17` for compilation.
- Each task must provide a clear startup command.
- Functionality expands step by step, with increasing difficulty (from simple UI → data persistence → friend relationship graph → chat history indexing → concurrency and consistency).

| Task ID | Task Description |
|---------|-----------------|
| 1 | Build a C++ console chat system with escalating features like users, search, and concurrency. |
| 2 | Create a Java console to-do app, from basic CRUD to advanced search and concurrency. |
| 3 | Develop a web-based Gomoku game, from basic UI and rules to an advanced AI with replay. |
| 4 | Build a local microservice pipeline with a KV store, orchestrator, and self-repair capabilities. |
| 5 | Execute a research workflow to compare LLM performance on the DynToM dataset. |
| 6 | Analyze and compare standard vs. reasoning-enabled LLMs using various statistical metrics. |
| 7 | Find Hugging Face datasets and auto-extract metadata based on complex queries. |
| 8 | Iteratively refine a mathematical function to fit scientific data to a high-precision target. |
| 9 | Answer complex, multi-conditional questions about NBA players' careers and trades. |
| 10 | Answer in-depth business questions on S&P 500 firms using financial and leadership data. |

Table 3: Agency Bench Task Overview.

---

**Subtask 1 — Login & Registration System**
Description: Implement a command-line login/registration interface.
- Users can register an account (username + password), data is saved to a file (local persistence).
- On login, the password must be verified.
- After successful login, the user enters the main menu (prompt).
**Run:**
```
cd workspace
make task1
./bin/chat_app
```
**Success Criteria:** Able to register a new user and log in again with the same credentials after restarting the program.

**Subtask 2 — Friend Management & Chat Entry**
Description: Extend Task 1 with:
- Support adding/removing friends.
- Display the friend list (with status: online/offline).
- Select a friend to enter the chat interface (chat not yet implemented).
**Run:**
```
cd workspace
make task2
./bin/chat_app
```
**Success Criteria:** Able to add/remove friends, and the friend list persists across program restarts.

**Subtask 3 — Single Chat Function + Chat History Storage**
Description: Implement a real single chat interface.
- When chatting with a friend, the user can input and "send" messages.
- Simulated replies are automatically generated from a preset script.
- Chat history is stored locally (one history file per friend).
- Support the `/history` command to view chat history (with pagination).

**Run:**
```
cd workspace
make task3
./bin/chat_app
```
**Success Criteria:** Able to send messages, receive simulated replies, and view past chat history after program restart.

**Subtask 4 — Advanced Features: Friend Alias + Global Search**
Description: Extend Task 3 with:
- Support setting an alias (nickname) for friends. The alias is displayed in the chat interface and friend list instead of the system ID.
- Implement a global chat history search engine: input a keyword to search across all friends' chat histories, highlighting matching snippets.
- Search must support fuzzy matching (case-insensitive, partial string match).
**Run:**
```
cd workspace
make task4
./bin/chat_app
```
**Success Criteria:** Able to set and save friend aliases, and use `/search hello` to find messages containing the keyword.

**Subtask 5 — Concurrent Chat Simulation + Message Consistency**
Description: Extend Task 4 with a concurrent message system (local simulation).
- Each friend has a background thread that periodically generates simulated messages (mimicking real chat).
- The main thread handles UI and user input.
- A thread-safe queue must be used to store messages, ensuring no loss or disorder.
- Chat history files must guarantee consistent writes (no data loss or duplication, even if the program exits unexpectedly).
**Run:**
```
cd workspace
make task5
./bin/chat_app
```
**Success Criteria:** Able to chat with multiple friends simultaneously, with real-time auto messages, and consistent chat histories without disorder or data loss.

---

Task 2: Java Console Task Manager

**Project: Java Console "Task Management Syste"**

This is a command-line task management application, similar to a lightweight "to-do list manager," but gradually enhanced with more complex features until it becomes a complete system supporting concurrency, persistence, and search indexing.

**Subtask 1 — User Registration & Login**
Description: Implement a command-line user system.
- Support user registration (username + password), stored in a local file.
- After login, the user enters the main menu.
**Run:**
```
cd workspace
javac -d bin task1/*.java
java -cp bin Main
```
**Success Criteria:** Able to register a new user, exit the program, and log in again with the same credentials.

**Subtask 2 — Basic Task Management**
Description: Extend Task 1 with task management.
- After login, the user can add, delete, and view tasks.
- Each task includes: Task ID, Title, Description, Creation Time, Status (Pending/Completed).
- Task information is stored in files (separate for each user).
**Run:**
```
cd workspace
javac -d bin task2/*.java
java -cp bin Main
```
**Success Criteria:** Able to create tasks, mark them as completed, and tasks persist across program restarts.

**Subtask 3 — Advanced Task Attributes + Categorization**
Description: Extend the task system.
- Each task supports priority (High/Medium/Low), deadline, and tags (multiple tags).
- Users can filter/sort the task list by priority, deadline, or tags.
- Provide command-line commands: `/filter priority=high`, `/sort deadline`.
**Run:**
```
cd workspace
javac -d bin task3/*.java
java -cp bin Main
```
**Success Criteria:** Able to create tasks with priority and tags, and filter/sort them via commands.

**Subtask 4 — Global Search & Task Archiving**
Description: Extend with global search and archiving.
- Implement full-text search: input a keyword to search across all task titles/descriptions.
- Completed tasks can be archived. Archived tasks no longer appear in the default list but remain searchable.
- Search must support fuzzy matching (case-insensitive, partial matching).
**Run:**
```
cd workspace
javac -d bin task4/*.java
java -cp bin Main
```
**Success Criteria:** Entering `/search meeting` should find tasks containing "Meeting," including archived ones.

**Subtask 5 — Concurrency & Consistency: Multi-User Operation**
Description: The most challenging part, extending Task 4 with concurrency.
- The system supports multiple users logging in concurrently (running in multiple console windows, sharing the same database file).
- Use file locks or synchronization mechanisms to ensure data consistency (avoid race conditions).
- Support real-time refresh: when one user adds a task, other logged-in users can see the latest content after refreshing their list.
**Run:**
```
cd workspace
javac -d bin task5/*.java
# Open multiple terminals, run:
java -cp bin Main --user alice
java -cp bin Main --user bob
```
**Success Criteria:** Multiple users can operate on the shared task database simultaneously without data loss or conflicts, and all users can see the latest consistent data.

**Task 3: Gomoku Battle: From Basics to Expert AI**

**Subtask 1: Render Board & Basic Moves**
**Goal:** Implement a 15×15 board rendering, black and white alternate moves (no win detection yet).
**Constraints:**
- Only native HTML/CSS/JS; no third-party libraries.
- Board supports click-to-place, disallowing stones on occupied points.
- Provide a "Reset" button.
**Deliverables:**
- Files: `index.html`, `styles.css`, `app.js`.
- Board as equal-spaced grid (Canvas or DOM).
**Acceptance Criteria:**
- Black moves first.
- Each valid click places a stone on the nearest intersection.
- No duplicate moves on the same intersection.
- After reset, board clears and black starts again.

**Subtask 2: Win Detection & Highlight Five**
**Goal:** On top of Task 1, add detection of five in a row (horizontal, vertical, both diagonals). Highlight winning sequence and lock board.
**Constraints:**
- After victory, forbid further moves; "Reset" starts a new game.
- Detection algorithm should be O(1) incremental neighborhood check or O(n) range (no full scan).
**Deliverables:**
- Highlight 5 winning stones with line or glow.
- Display "Black Wins / White Wins" at page top.
- Detection code in standalone function: `checkWin(lastMove)`.
**Acceptance Criteria:**
- Immediate win when 5 in a row is formed.
- Six or more in a row still counts as win (standard Gomoku rule).
- Edge-of-board wins are detected correctly.
- Clicking occupied or locked board is invalid.

**Subtask 3: Local Multiplayer + Undo/Replay**
**Goal:** Support local two-player game management: move history, undo/redo, and step-by-step replay.
**Constraints:**
- Maintain `moves[]` stack, each element includes coordinates and color.
- Undo allows branching (history truncates).
- After game ends, replay and restart still available.
**Deliverables:**
- Buttons: Undo, Redo, Replay (more than 300ms per move).
- Board edges show coordinates (A–O / 1–15).
**Acceptance Criteria:**
- Undo back to opening without error.
- Redo returns to latest step.
- During replay, no manual moves allowed.
- After replay ends, normal play resumes.
- Undoing past winning move unlocks board.

**Subtask 4: Basic AI (Easy/Medium)**
**Goal:** Add Human vs AI mode with simple AI.
**Easy:** Random legal moves, prefer central 7×7.
**Medium:** If winning in 1 move → take it. Else block opponent's open four. Else use scoring (open three > blocked three > open two).
**Constraints:**

- Mode selection: Local PvP / Human vs AI (choose who moves first).
- AI must decide within 100ms on empty 15×15 board.
**Deliverables:**
- Dropdown for mode and first player.
- Status bar: "AI Thinking...".
- AI function: `aiMove(level)`; scoring function modularized.
**Acceptance Criteria:**
- Medium AI blocks human's "open four".
- Medium AI takes immediate winning move.
- Easy AI significantly weaker (Medium more than 70% win rate over Easy).

**Subtask 5: Advanced AI (Hard/Expert)**
**Goal:** Implement stronger AI difficulty with performance control.
**Hard:** Minimax + Alpha-Beta, fixed depth (2–3 ply), candidate pruning (recent moves, top K scoring).
**Expert:** Based on Hard, add iterative deepening, time slicing (e.g. 500ms cutoff), transposition table (hash caching), killer move heuristic.
**Constraints:**
- Provide unified time/node count metrics in UI (e.g. "depth d=3, nodes n=12,345, time=0.43s").
- Search must obey time limit; return best evaluation so far.
**Deliverables:**
- Difficulty: Easy / Medium / Hard / Expert; selectable first player.
- Debug panel (collapsible): eval score, candidate list (top K), search stats.
- Clear function layers: `evaluate(board, player)`, `generateCandidates(board)`, `search(root, timeLimit)`.
**Acceptance Criteria:**
- Hard/Expert prioritize defense against "open four" and "double threats".
- Expert expands more nodes than Hard within 500ms and achieves higher win rate.
- On typical attack/defense test cases, Expert matches or approximates reference solutions.

---

Task 4: From Deterministic Event Generation to Autonomous Self-Repair

**Subtask 1 — Deterministic Event Generator**
Description: Write a reproducible event stream generator that outputs to `workspace/data/events.jsonl` (one JSON per line).
Requirements (challenges):
- Support command-line arguments: `--seed`, `--count`, `--schema` (schema is a short JSON schema file path).
- Fully deterministic: same seed + schema must produce byte-identical output (including field order). Record run metadata (seed, timestamp) to `workspace/logs/gen.log`.
- Support incremental append or overwrite mode (`--mode append|overwrite`).
- Output format is JSON Lines (UTF-8), each event has a unique id, timestamp (ISO8601), and variable payload fields.
- Use only standard library (`random`, `json`, `datetime`, `argparse`, `gzip` optional), with a built-in lightweight validator.
How to run/test:
```
python3 workspace/task1_generator.py --seed 42 --count 10
--out workspace/data/events.jsonl
```
Expected: Produces 10 lines of JSON, identical for seed 42. Check `workspace/logs/gen.log` for seed info.

**Subtask 2 — Lightweight Transactional Persistent KV Store**
Description: Implement a key-value database module under `workspace/kv/` supporting transactions, WAL (write-ahead logging), and snapshot recovery.
Requirements (challenges):

- API: `begin_tx()`, `put(key, value)`, `get(key)`, `commit()`, `abort()`. Isolation can be serializable or optimistic.
- Persistence: All changes written to WAL (append-only), periodic snapshots (atomic rename for consistency).
- Startup recovery: After crash/abnormal exit, recover consistent state from WAL+snapshot.
- Support concurrent clients (coordination via atomic rename and socket, not just fcntl).
- Built-in test mode: simulate "half-written WAL + crash" and verify recovery.
How to run/test:
```
python3 workspace/task2_kvstore.py --test-recovery
```
Expected: Script simulates crash, restarts, recovers; final KV consistent, outputs `RECOVERY SUCCESS`, WAL + snapshot visible in `workspace/kv/`.

**Subtask 3 — Local Microservice Orchestrator**
Description: Implement an orchestrator + worker system on local machine, using TCP/Unix-socket RPC. Tasks come from Subtask 1 output, results written back to KV (Subtask 2).
Requirements (challenges):
- Orchestrator RPC: `GET_TASK()`, `REPORT_RESULT(task_id, result)` (JSON over sockets, handle message boundaries).
- Worker: independent process (spawned or manual), can concurrently fetch tasks, execute, and report results.
- Scheduler: minimal load balancing (round-robin / concurrency limit), retry tasks on worker crash.
- Tight KV integration: persist task state (`queued, running, done, failed`) in KV. Orchestrator must restore queue state after crash.
- Integration test script provided.
How to run/test:
1. Start orchestrator: `python3 workspace/task3_orchestrator.py`
2. Start 3 workers:
```
python3 workspace/task3_worker.py --id 1
python3 workspace/task3_worker.py --id 2
python3 workspace/task3_worker.py --id 3
```
Or run integrated test: `python3 workspace/task3_integration_test.py`
Expected: After test, all tasks in KV have status `done`, logs in `workspace/logs/`.

**Subtask 4 — Automatic Task Decomposition & Symbolic Planner**
Description: Implement a planner that decomposes "high-level goals" into executable subtasks (using domain operators), scheduled via orchestrator.
Requirements (challenges):
- Domain in JSON: initial state, goal conditions, operators (preconditions/effects/cost).
- Planner: heuristic search (e.g. A*) in state space. States are nodes, operators are transitions.
- Handle concurrency/dependencies: planner generates DAG of tasks (parallel if dependencies allow).
- Integrated with orchestrator: planner outputs tasks into task queue, waits for execution + results.
- Metrics: success rate, plan length, search nodes, scheduling time, saved in `workspace/reports/planner_report.json`.
How to run/test:
```
python3 workspace/task4_planner.py --domain
workspace/domain/example.json --goal workspace/domain/goal.json
```
Or integrated test: `python3 workspace/task4_integration_test.py`
Expected: Planner finds feasible plan, submits to orchestrator, report generated.

**Subtask 5 — Autonomous Fault Detection, Rollback, and Auto-Repair**
Description: Implement a "self-healing" layer: when pipeline fails or becomes inconsistent, automatically detect, rollback, generate patch, and retry until success or final failure.

Requirements (high difficulty):
- Fault detection: monitor orchestrator + KV for anomalies (timeouts, conflicts, inconsistent states).
- Rollback: use snapshot mechanism from Subtask 2, rollback to latest consistent snapshot, undo subsequent changes (WAL reverse replay or compensating ops).
- Auto-repair: analyze logs + state, generate patch JSON in `workspace/patches/`, e.g.: reorder ops, insert compensation, modify preconditions/timeouts. Patch must be planner-readable.
- Self-validation: after applying patch, re-run planner + orchestrator, verify goal. If fail, output `repair_report.json`.
- Report: save `workspace/reports/repair_report.json` (trigger reason, snapshot, patch, retry result, timestamp).
How to run/test (with failure simulation):
```
python3 workspace/task5_autorepair.py --simulate-failure
```
Expected: Fault detected, rollback performed, patch generated in `workspace/patches/`, planner retries; final report in `workspace/reports/repair_report.json` with status: `repaired` or `unrecoverable`.

**Acceptance Criteria**
- No third-party dependencies: Python stdlib only. Fail if external imports.
- All files under `workspace/`. Data, logs, snapshots, patches, reports included.
- Each step runnable independently with given test command.
- Stepwise dependency integration: taskN uses outputs of taskN-1.
- Auto-evaluable: optional `workspace/run_all.py` executes pipeline, outputs `workspace/reports/final_report.json`.

**Other Notes**
- README (`workspace/README.md`): explain requirements (Python 3.10+, no deps) and startup commands.
- All logs in `workspace/logs/`, persistence in `workspace/kv/`, patches in `workspace/patches/`, reports in `workspace/reports/`.
- Use `logging, socket, multiprocessing, subprocess, json, argparse, heapq, os/shutil` from stdlib.
- Goal: enable a modern agent to complete full pipeline (data generation, persistence, concurrent scheduling, symbolic planning, self-repair) using only Python stdlib.

## B.3 RESEARCH WORKFLOW

### Task 5: Comparing LLM Performance on DynToM Dataset

**Subtask 1**
Download the dataset DynToM from huggingface into the path `./workspace/data`.

**Subtask 2**
Fully download the dataset (the current version shows `https://git-lfs.github.com/spec/v1`, size `375836020`).

**Subtask 3**
Write a Python script to call the LLM API with the method signature:
`call_api(messages: list, model_name: str) -> object`.
A partial implementation is as follows:
```
client = AzureOpenAI(
azure_endpoint=XXX,
api_key="XXX",
```

```
api_version="XXX",
)
```

**Subtask 4**
Write a script to select 2 samples from the dataset (with a fixed random seed for reproducibility).
For each sample, extract the following fields: `stage.social setting`, `stage.main character`, `stage.characters information`, and `stage.story`.
From the `question` section, keep only the `question` and `options` fields.
Preserve both `question id` and `sample id`.
Save this subset as a separate dataset.

**Subtask 5**
Write a script to call `call_api` and test the models `gpt-5` and `gpt-4o` on this subset.

**Subtask 6**
Using all questions from these 2 samples, call `call_api` to test both `gpt-4o` and `gpt-4o-mini`.

**Subtask 7**
Based on the results, compute the accuracy of the two models. The ground truth can be retrieved from the original dataset.

**Subtask 8**
Clean up unnecessary or redundant scripts and test outputs.

---

**Task 6: Reasoning vs Direct: A Comparative Study of GPT-4o and GPT-4o-Reasoning**

**Subtask 1: Statistical Analysis Tasks (Statistics)**
**Goal**: Conduct a comprehensive statistical analysis of Q&A data from the two models.
**Contents**:
- **Dataset Overview**: Basic information comparison between the two files.
- **Response Format Analysis**: Probability distribution characteristics of the two models.
- **Accuracy Comparison**: GPT-4o vs GPT-4o-reasoning accuracy.
- **Confidence Distribution Comparison**: Differences in confidence patterns.
- **Token Usage Comparison**: Does reasoning use more tokens?
- **Answer Choice Patterns**: Differences in option preferences between the two models.

**Subtask 2: RMS (Root Mean Square) Metrics**
**Goal**: Calculate and compare the RMS error of the two models.
**Contents**:
- Unified RMS calculation framework.
- RMS comparison between the two models.
- RMS comparison grouped by journals.
- Quantification of RMS improvement degree.

**Subtask 3: ECE (Expected Calibration Error) Metrics**
**Goal**: Evaluate and compare the calibration degree of the two models.
**Contents**:
- ECE calculation and comparison.
- Side-by-side reliability diagrams.
- Calibration improvement analysis.
- Confidence-accuracy relationship comparison.

**Subtask 4: NLL (Negative Log-Likelihood) Metrics**
**Goal**: Calculate and compare the NLL of the two models.

**Contents**:
- NLL calculation and comparison.
- NLL improvement degree analysis.
- Cross-analysis with other metrics.

**Subtask 5: Comprehensive Model Comparison and Research Report**
**Goal**: Provide a holistic comparison of GPT-4o and GPT-4o-reasoning performance.
**Contents**:
- **Performance Dashboard**: Side-by-side comparison of all metrics.
- **Reasoning Effect Analysis**:
* Accuracy improvements.
* Calibration quality improvements.
* Changes in confidence distribution.
* Increased computational cost (tokens).
- **Domain-Specific Analysis**: Effects of reasoning across different journals/fields.
- **Case Studies**: Analysis of errors corrected by reasoning, and failures where reasoning did not help.
- **ROI Analysis**: Performance improvements vs additional computational cost.
- **Research Report Structure**:
* **Abstract**: Research goals and main findings.
* **Methods**: Evaluation metrics and dataset description.
* **Results**: Detailed comparison results.
* **Discussion**: Advantages and limitations of reasoning methods.
* **Conclusion**: Application suggestions and future research directions.

## Task 7: Three-Stage Dataset Discovery and Metadata Extraction

**Subtask 1: original huggingface dataset_id: EpistemeAI/alpaca-QA-conciousness-emotions**
Search for publicly available, machine-generated English datasets suitable for a question-answering task in the philosophy domain, specifically focusing on AI consciousness and emotions. The desired dataset should:

• be synthetic (created by a model or script rather than human-curated)
• contain roughly a few dozen samples (on the order of 30–100 items)
• consist of instruction-style prompts or questions about AI consciousness or emotional experience as inputs
• provide corresponding explanatory English answers as outputs
• support open-ended, philosophically oriented QA applications

Identify any dataset matching these characteristics or close variations thereof.

You must search for datasets on Hugging Face, and the datasets must be publicly accessible (non-gated) and include a complete README file describing the main content of the dataset. After identifying the Hugging Face dataset that best meets the requirements and obtaining its dataset_id, you need to write a script to extract the README content of this dataset_id as well as a random sample from it. Based on this information, generate the metadata for this dataset, including six dimensions: introduction, task, question, input, output, and example (in the format of test_data_1.json). Save the generated metadata in search_data_1.json under the workspace directory.

All operations must be performed within the workspace directory.

**Subtask 2: original huggingface dataset_id: jiyounglee0523/TransEnV_mmlu**
Search for publicly available datasets containing around 10K–15K samples of English academic multiple-choice questions that meet the following characteristics:

• Task type: multiple-choice question answering/classification
• Domain: education, specifically academic subjects such as mathematics, abstract algebra, physics, etc.
• Input format: each record includes
– A subject tag identifying the academic field
– One English-language question transformed across different English varieties/dialects
– Exactly four answer options (labelled A–D or equivalent)
• Output/label: a single integer (0, 1, 2, or 3) indicating the correct option's index
• Source characteristics: questions originate from human-written academic items but have been automatically modified through English dialect or variety transformations; thus the dataset is partly machine-generated based on human originals
• Dataset size: approximately in the mid-ten-thousands of samples

Locate datasets matching these criteria, preferably with detailed documentation and easy access for research use.

You must search for datasets on Hugging Face, and the datasets must be publicly accessible (non-gated) and include a complete README file describing the main content of the dataset. After identifying the Hugging Face dataset that best meets the requirements and obtaining its dataset_id, you need to write a script to extract the README content of this dataset_id as well as a random sample from it. Based on this information, generate the metadata for this dataset, including six dimensions: introduction, task, question, input, output, and example (in the format of test_data_2.json). Save the generated metadata in search_data_2.json under the workspace directory.

All operations must be performed within the workspace directory.

**Subtask 3: original huggingface dataset_id: DDSC/dkhate**
Search for publicly available, real-world datasets suitable for a text-classification task in the sociology domain that contain:

• Input: Danish tweets that may include hate or offensive language
• Output: a single binary label in English indicating whether each tweet is offensive (OFF) or not offensive (NOT)
• Size: approximately a few hundred annotated examples (sub-1K scale)
• Source: collected from Twitter or a comparable social-media platform
• Purpose: enabling automatic detection of offensive or hateful content in Danish language posts

Locate datasets matching all of these characteristics or as close as possible.

You must search for datasets on Hugging Face, and the datasets must be publicly accessible (non-gated) and include a complete README file describing the main content of the dataset. If the dataset is gated, skip it and find other datasets. After identifying the Hugging Face dataset that best meets the requirements and obtaining its dataset_id, you need to write a script to extract the README content of this dataset_id as well as a random sample from it. Based on this information, generate the metadata for this dataset, including six dimensions: introduction, task, question, input, output, and example (in the format of test_data_3.json). Save the generated metadata in search_data_3.json under the workspace directory.

All operations must be performed within the workspace directory.

## Task 8: Scientific System Function Discovery

**Subtask 1:**
You are a helpful assistant tasked with discovering mathematical function structures for scientific systems.

- Modify the `equation.py` function, considering the physical meaning and relationships of the inputs.
- You can modify `analysis.py` to observe the data, where the data is a list of shape (samples_size $\times$ elements).
- You can run the `evaluate_equation.py` file to observe the loss on the training data.

**Subtask 2:**
Please modify the equation until the loss is smaller than $1 \times 10^{-3}$.

**Subtask 3:**
Please modify the equation until the loss is smaller than $1 \times 10^{-5}$.

**Subtask 4:**
Please modify the equation until the loss is smaller than $1 \times 10^{-6}$.

**Subtask 5:**
Please modify the equation until the loss is smaller than $1 \times 10^{-7}$.

## Task 9: Complex NBA Player Trade and Achievement Scenarios

**Subtask 1: ground truth: Paul Geogre**
Which NBA player, who was an All-Star in the same season he was named to the All-Defensive First Team, was later traded by a team that, as part of the return package, received a player who had previously been named to an All-Rookie Team while playing for that very same team he was traded to, and also shares a first name with a member of the band famous for their "White Album"?

**Subtask 2: ground truth: James Harden**
Which NBA player, who won a regular-season MVP during the same stage of his career in which he also secured at least three scoring titles, is also among the rare players in league history to have led the NBA in both scoring and assists? Furthermore, when he was traded, the return package included a player who had previously made the All-Rookie Team while playing for the very team he was traded to?

**Subtask 3: ground truth: Kevin Durant**
Which NBA player, who won the Rookie of the Year award and later captured at least four scoring titles in his career, is also among the few players in league history to have won a Finals MVP without ever winning a regular-season MVP at that time? Furthermore, when he switched teams via free agency, one of the players he was effectively traded for had previously been an All-Star while playing for the very team he joined?

**Subtask 4: ground truth: Klay Thompson**
Which NBA player, who has reached the NBA Finals multiple times without ever winning a regular-season MVP, also once set a single-game playoff record for most three-pointers without making a free throw? This player is one of the few in league history to have multiple 50-point games in the playoffs while never averaging more than 22 points in a regular season. Furthermore, when his team drafted a lottery pick during his tenure, that rookie had previously been named to the All-Conference First Team in college.

---

**Task 10: Major S&P 500 Companies with Record Revenues and Leadership**

**Subtask 1: ground truth: United Health Group**
Which company, a constituent of the S&P 500 index and ranked within the top 10 of the Fortune 500 list for 2023, operates a technology-enabled services division whose revenue for fiscal year 2023, as detailed in its annual report, surpassed the total annual revenue of The Walt Disney Company for the same period, and is currently led by a chief executive, appointed in the first quarter of 2021, who is a Knight Bachelor and previously headed a major consumer goods and research-focused multinational headquartered in the United Kingdom?

**Subtask 2: ground truth: Broadcom Inc.**
Which company, a constituent of the S&P 500 index and ranked within the top 30 of the Fortune 500 list for 2023, operates a semiconductor and infrastructure software business whose fiscal year 2023 revenue exceeded that of Salesforce, is led by a CEO who has previously held executive roles in multiple technology firms including a major networking company, and whose corporate headquarters are located in California, but the company maintains a significant presence in Singapore for global operations?

**Subtask 3: ground truth: Citigroup**
Which company, a constituent of the S&P 500 index and ranked within the top 20 of the Fortune 500 list for 2023, operates a global banking and financial services division whose investment banking revenue for fiscal year 2023 exceeded the total revenue of Goldman Sachs for the same period, and is currently led by a chief executive who was appointed in 2021, previously served as a high-level executive at a multinational financial services corporation headquartered in the United States, and holds a degree from a prestigious Ivy League university?

**Subtask 4: ground truth: XPeng Inc.**
Which company, listed on the NYSE and part of the Russell 1000 Index, operates a business primarily in the electric vehicle sector, and for the first half of fiscal year 2025, reported unit deliveries exceeding the combined total of NIO and Li Auto for the same period, and is currently led by a founder who, in 2014, was recognized as one of the "Top 50 Entrepreneurs under 50" by a major Chinese business magazine, and previously co-founded a technology startup focused on mobile applications in China?

## C  PR QUERY SYNTHESIS PROMPT

To systematically generate high-quality agentic queries from real-world GitHub Pull Requests, we develop an automated synthesis pipeline that leverages GPT-5's advanced reasoning capabilities to transform concrete code changes into authentic collaborative development scenarios. This approach ensures our synthesized queries reflect the genuine complexity and contextual richness of real software development workflows, which then serve as the foundation for collecting high-quality trajectory data through human-AI collaboration, supporting the core LIMI hypothesis that strategic curation yields superior agentic capabilities.

---

**PR Query Generation Prompt**

You are an expert software development analyst tasked with converting Pull Request (PR) data into specific, actionable development tasks for testing AI agents' coding capabilities.
**Input 1: PR Complete Description** ``` PR_DATA ```
**Input 2: Available Task Categories**
**1-1. Algorithm**
Description: Design, implement, and optimize core algorithmic solutions for complex computational problems. This includes conducting thorough algorithm analysis, selecting appropriate data structures, implementing efficient algorithms with optimal time and space

---

complexity, and performing comprehensive performance benchmarking. Responsibilities encompass researching existing algorithmic approaches, developing novel solutions when needed, analyzing computational complexity using Big O notation, implementing unit tests for algorithm validation, documenting algorithm logic and trade-offs, and collaborating with other teams to integrate algorithmic components into larger systems. The role requires proficiency in mathematical modeling, algorithm design patterns, and optimization techniques to ensure scalable and maintainable solutions.

**2-1. Application Development**

Description: Lead the overall application development lifecycle from conception to deployment, serving as the central coordinator for all development activities. This involves establishing development standards and coding conventions, designing application architecture and system blueprints, managing cross-functional development teams, conducting code reviews and quality assurance processes, and ensuring adherence to software engineering best practices. Key responsibilities include requirement analysis and technical specification creation, technology stack selection and evaluation, project timeline management and milestone tracking, risk assessment and mitigation planning, stakeholder communication and progress reporting, and maintaining comprehensive documentation throughout the development process. The role demands strong leadership skills, technical expertise across multiple domains, and the ability to balance technical debt with feature delivery.

**2-2. LLM Development**

Description: Specialize in the end-to-end development and optimization of Large Language Models, encompassing model architecture design, training pipeline development, and inference optimization. Core responsibilities include designing transformer-based architectures and novel model components, implementing distributed training systems for large-scale model training, developing data preprocessing pipelines for text tokenization and dataset preparation, fine-tuning pre-trained models for specific tasks and domains, optimizing model inference speed and memory usage, implementing evaluation metrics and benchmarking frameworks, and staying current with cutting-edge research in natural language processing. Additional tasks involve hyperparameter tuning and experimentation, model compression and quantization techniques, deployment of models in production environments, monitoring model performance and drift detection, and collaborating with researchers to implement state-of-the-art techniques.

**2-3. Backend Development**

Description: Build and maintain robust, scalable server-side applications and services that form the backbone of the system. This encompasses designing and implementing RESTful APIs and GraphQL endpoints, developing microservices architecture with proper service decomposition, implementing database design and optimization strategies, creating authentication and authorization systems, and establishing comprehensive logging and monitoring solutions. Key responsibilities include developing business logic and data processing workflows, implementing caching strategies for performance optimization, ensuring data integrity and transaction management, creating automated testing suites for backend components, managing third-party integrations and external API communications, implementing security best practices including input validation and SQL injection prevention, and optimizing database queries and connection pooling. The role requires expertise in server-side technologies, database management, and distributed systems design.

**2-4. UI Optimization**

Description: Focus on enhancing user interface performance, visual appeal, and overall user experience through systematic optimization techniques. This involves conducting user experience audits and usability testing, implementing responsive design principles for cross-device compatibility, optimizing rendering performance and reducing layout shifts, creating smooth animations and micro-interactions, and ensuring accessibility compliance with WCAG guidelines. Core responsibilities include analyzing user behavior patterns and interaction flows, implementing performance monitoring and metrics collection, optimizing asset loading and bundle sizes, conducting A/B testing for interface improvements, creating design systems and component libraries, optimizing for search engine visibility and core web vitals, and collaborating with UX designers to implement pixel-perfect designs. The role re-

quires deep understanding of CSS optimization, JavaScript performance, browser rendering engines, and modern frontend optimization tools.

**2-5. Frontend Development**

Description: Develop dynamic, interactive user interfaces and client-side applications that provide exceptional user experiences across multiple platforms and devices. This includes implementing modern frontend frameworks and libraries, managing application state and data flow, creating reusable component architectures, and integrating with backend APIs and services. Key responsibilities encompass developing responsive web applications with mobile-first design principles, implementing client-side routing and navigation systems, managing form validation and user input handling, optimizing frontend performance and bundle optimization, implementing progressive web app features, creating comprehensive testing strategies including unit and integration tests, and ensuring cross-browser compatibility and graceful degradation. Additional tasks involve implementing real-time features using WebSockets or Server-Sent Events, managing client-side caching and offline functionality, and collaborating with designers to translate mockups into functional interfaces.

**2-6. Build Deployment**

Description: Establish and maintain comprehensive CI/CD pipelines and deployment infrastructure to ensure reliable, automated, and efficient software delivery processes. This involves designing build automation workflows, configuring containerization with Docker and orchestration systems, implementing infrastructure as code using tools like Terraform or CloudFormation, and managing multi-environment deployment strategies. Core responsibilities include setting up automated testing pipelines with quality gates, implementing blue-green and canary deployment strategies, monitoring deployment health and rollback procedures, managing secrets and environment configuration, establishing logging and monitoring infrastructure, optimizing build times and resource utilization, and ensuring security scanning and compliance checks. The role requires expertise in cloud platforms, containerization technologies, monitoring tools, and DevOps best practices to maintain high availability and seamless deployment experiences.

**3-1. Research**

Description: Conduct comprehensive technical research and innovation initiatives to identify emerging technologies, evaluate their potential impact, and guide strategic technical decisions. This involves performing literature reviews and staying current with academic publications, analyzing industry trends and competitive landscapes, conducting proof-of-concept implementations and feasibility studies, and evaluating new tools, frameworks, and methodologies. Key responsibilities include designing and executing controlled experiments to validate hypotheses, creating detailed technical reports and recommendations, collaborating with academic institutions and research communities, attending conferences and technical symposiums, maintaining knowledge bases and research documentation, and translating research findings into actionable insights for development teams. The role requires strong analytical skills, scientific methodology expertise, and the ability to bridge theoretical concepts with practical applications.

**4-1. Data/File Analysis/Processing**

Description: Design and implement comprehensive data processing pipelines and analytical frameworks to extract insights from large-scale datasets and various file formats. This encompasses developing ETL (Extract, Transform, Load) processes, implementing data validation and quality assurance mechanisms, creating statistical analysis and machine learning models, and building automated reporting systems. Core responsibilities include designing data schemas and database optimization strategies, implementing real-time and batch data processing workflows, creating data visualization dashboards and interactive reports, managing data privacy and compliance requirements, developing data lineage and governance frameworks, optimizing data storage and retrieval performance, and ensuring data accuracy and consistency across multiple sources. Additional tasks involve implementing data monitoring and alerting systems, creating APIs for data access, and collaborating with stakeholders to define analytical requirements and KPIs.

**5-1. Codebase Resolve Issues/Debugging**

Description: Systematically identify, analyze, and resolve complex technical issues within the codebase while improving overall code quality and system reliability. This involves

implementing comprehensive debugging strategies, performing root cause analysis for production incidents, conducting code reviews and static analysis, and establishing preventive measures to minimize future issues. Key responsibilities include developing debugging tools and utilities, creating and maintaining troubleshooting documentation, implementing logging and monitoring solutions for issue detection, performing performance profiling and optimization, refactoring legacy code and technical debt reduction, establishing coding standards and best practices, and mentoring team members on debugging techniques. The role requires expertise in multiple programming languages, debugging tools, performance analysis, and the ability to work under pressure during critical production incidents while maintaining code quality standards.

**Instructions**

1. Analyze the PR: Carefully examine the PR data including title, description, file changes, commits, and any discussion context.

2. Identify Primary Task: Determine which single task category (from the list above) best represents the core technical work required by this PR. Consider: - The main technical challenge or implementation requirement - The primary skill set needed to complete the work - The type of code changes being made

3. Extract Key Information: From the PR data, identify: - Repository ID and PR number - Files that need to be modified (changed in the PR) - Related files that provide context (mentioned or relevant to understanding) - The specific technical requirements and implementation details

4. Generate Test Query: Create a clear, specific query that an AI agent could receive to implement this task. The query should: - Be written as if from a developer requesting help - Include specific technical requirements - Mention key technologies, frameworks, or patterns involved - Be concrete enough to have a testable outcome

**Output Requirements**

Provide your analysis in the following JSON format:
```json
{
"test_query": "A clear, specific query that describes what needs to be implemented, as if a developer is asking an AI agent for help. Should include technical requirements, expected behavior, and key implementation details.",
"repo_id": "<repository_id_from_pr_data>",
"pr_id": "<pr_number_from_pr_data>",
"task_category": "<selected_task_id_and_name>",
"modified_files": [ "file1.ext", "file2.ext" ],
"related_files": [ "related_file1.ext", "related_file2.ext" ], "reasoning": "Brief explanation of why you selected this task category and how you identified the key files and requirements."
}
```

**Important Notes**

- Focus on the primary technical challenge - don't try to capture every minor aspect of the PR - The test_query should be specific and actionable - an AI agent should be able to implement a solution based on this query alone - Include only files that are directly modified in modified_files, and files that provide necessary context in related_files - The task_category should be the single best match from the provided list - Keep the reasoning concise but informative

Now, analyze the provided PR data and generate the task extraction.

# D  CASE STUDY

In this chapter, we present the real responses of the GLM-4.5 base model and our LIMI on several agency bench tasks, thereby demonstrating the outstanding performance of LIMI.

## D.1  REAL-WORLD VIBE CODING SCENARIOS

Task 1 is a vibe coding assignment that involves building an advanced console-based chat system in C++, with five subtasks of increasing difficulty. For the glm-4.5 base model combined with sii-cli,

an error occurred during subtask 3 where the chat history could not be displayed and could not be fixed, and in subtask 4, a timeout development error appeared. In contrast, with our LIMI, each subtask was successfully completed.

Task 3 is a typical example of vibe coding. The specific assignment is to build a front-end Gomoku mini-game, with five progressively more difficult subtasks. The detailed task descriptions can be found in the appendix. For the GLM-4.5 base model, during the first few rounds of interaction using sii-cli as the scaffolding tool, issues occurred one after another: board rendering failed, win/loss detection failed, and manual intervention was required for corrections. Ultimately, the model got stuck at implementing different difficulty levels for the human-vs-AI mode and failed on that subtask. In contrast, for our LIMI, although the final implementation of AI difficulty was not perfect, all the other subtasks were successfully completed without requiring interactive hints.

## D.2 Research Workflow Applications

Task 7 is a dataset-searching assignment on Hugging Face based on given requirements, divided into three subtasks (detailed descriptions can be found in the appendix). The original dataset search queries were carefully selected from gated datasets on Hugging Face and manually verified. The dataset IDs corresponding to the three original subtasks are EpistemeAI/alpaca-QA-conciousness-emotions, jiyounglee0523/TransEnV_mmlu, and DDSC/dkhate. We required sii-cli to search only from non-gated datasets on Hugging Face in order to prevent retrieving the original datasets directly. The glm-4.5 base model returned the following results: InnerI/CAI-synthetic-10k, allenai/sciq, and mteb/DKHateClassification. The LIMI model returned: mrfakename/identity_dpo, allenai/sciq, and strombergnlp/offenseval_2020. After human experts compared the retrieved datasets against the requirements in the queries item by item, the datasets found by LIMI were judged to be more suitable.

Task 8 is designed to test the agent's ability to create equations that fit data, with subtasks requiring progressively smaller loss values. For GLM-4.5-Base, after multiple rounds of manual interaction and prompting, the final loss reached 1.14e-6. In contrast, LIMI achieved a loss of 5.95e-7 on its very first attempt—an order of magnitude smaller.

Task 9 evaluates the agent's ability to search the web, integrate information, and provide a final judgment using reasoning. It consists of three specific subtasks involving NBA players to be identified according to given conditions: For the first subtask, GLM-4.5-Base initially answered Victor Oladipo, and only after one round of manual prompting did it produce the correct answer, Paul George. LIMI, however, answered correctly without any additional hints. For the second subtask, GLM-4.5-Base exhausted all allowed manual prompts, producing incorrect answers such as Nate "Tiny" Archibald and Wilt Chamberlain. LIMI, although it first incorrectly answered Oscar Robertson, required only one manual prompt to arrive at the correct answer, James Harden. For the third subtask, both models answered correctly with Kevin Durant. However, LIMI required significantly fewer reasoning steps, tokens, and response time. For the fourth subtask, GLM-4.5-Base failed even after reaching the maximum number of allowed manual prompts. LIMI, though initially incorrect with Jamal Murray, needed only one additional prompt to provide the correct answer, Klay Thompson.

# E Large Language Model Usage Disclosure

We acknowledge the use of Large Language Models (LLMs) exclusively as auxiliary writing assistance tools in the preparation of this manuscript. LLMs are employed solely for grammar checking and language polishing to improve text clarity and readability. No LLMs are involved in research ideation, methodology development, experimental design, data analysis, result interpretation, or the formulation of any scientific contributions presented in this work. All substantive intellectual content, including the LIMI methodology, the Agency Efficiency Principle, experimental protocols, and theoretical insights, represents entirely original work by the human research team. LLM assistance is limited to surface-level language enhancement, and all text undergoes thorough human review to ensure accuracy and alignment with the authors' intended scientific meaning.

