# OpenReview forum: "LIMI: Less is More for Agency"
_ICLR.cc/2026/Conference — Submitted to ICLR 2026_

### Official Review · Reviewer_dqPa · 2025-10-25

**Soundness:** 2
**Presentation:** 2
**Contribution:** 2
**Rating:** 2
**Confidence:** 4

**Summary:**

This paper introduces a concept called LIMI (Less Is More for Intelligent Agency) for training agentic large language models using a very small, high-quality dataset of only 78 curated trajectories. The authors argue that “agency” of AI, which is defined as the ability to set goals, plan, act, and reflect, can emerge from a handful of strategically selected examples rather than massive data accumulation. They claim that a small number of “archetypal” agentic examples encode sufficient behavioral structure to elicit autonomous behavior.

**Strengths:**

1. Clear motivation: The claim that autonomy could emerge from a strategically curated dataset is intellectually provocative, which is a natural extension to previous papers such as LIMA.
2. Interesting observation: The observation that LLMs can be improved significantly in vide coding and research workflows with a small amount of data is interesting, which can potentially benefit research in automating vibe coding and research workflows with LLMs.

**Weaknesses:**

1. Conceptual Clarity:  The paper's definition of "agency", which is the capacity to set goals, plan, act, and adapt, is conceptually intuitive but not formally defined or measurable. The authors fail to provide any operational test distinguishing "agentic" from "non-agentic" behavior. The experimental dataset covers only two domains, vibe coding and research workflows, which significantly limits generality. Although vibe coding and research workflows are popular applications of LLMs, such definitions about "agency" is inherently subjective and ambiguous.
2. Technical Novelty: The paper's tone is overstating its technical novelty. Similar data-efficiency ideas have been extensively explored by previous works in LIMA or LIMO. Only changing the contexts without discussing the fundamental difference provide very limited insights into LLMs post-training. In addition, while the experiments are only conducted with LLMs, the authors are claiming some findings around general AI, also without any clarification about the difference.
3. Methodological Rigor: The authors claim that LLMs can be improved significantly through training with only 78 examples. However, the data-efficiency comparison (78 vs. 10k) is striking but potentially confounded by uncontrolled variables, for which the authors provide very limited explanations. While the curation process is described in detail, it is heavily subjective (three PhD annotators rating examples). There is also no ablation verifying whether LIMI’s gain stems from pure curation quality, domain specificity, or inherent properties of the data. Thus, I strongly feel the claim that “adding more data hurts agency” lacks systematic testing.

**Questions:**

1. How was the 78-example subset chosen beyond human scoring, was there quantitative validation of “agency density”? Are there any ideas that people can actually use your concepts to do strategic data curation?
1. How exactly is agency defined and measured? What is the fundamental difference between your "agency" and typical reasoning and alignment tasks? What is the connection of your "agency" with established theories or concepts (e.g., theory of agencies)? Why are vibe coding and research workflows sufficient to represent such tasks?

---

> ### Author Response · Authors · 2025-12-02
> **Rebuttal to Reviewer dqPa (1/3)**
>
> We sincerely thank Reviewer dqPa for the thoughtful feedback. We address each concern below with additional clarifications and new experimental evidence.
>
> ---
>
> > W1: The paper fails to provide an operational test distinguishing "agentic" from "non-agentic" behavior.
>
> > Q2: What is the fundamental difference between your "agency" and typical reasoning and alignment tasks?
>
> We respectfully disagree that our definition lacks operational clarity. In Section 2 (line 58), we provide a formal definition aligned with established literature [1,2]:
>
> *"Agency as the emergent capacity of AI systems to function as autonomous agents—actively discovering problems, formulating hypotheses, and executing solutions through self-directed engagement with environments and tools."*
>
> **Operational distinctions from reasoning/alignment tasks:**
>
> Agentic systems differ fundamentally from reasoning/alignment tasks through three key dimensions: they interact with real-world environments and tools rather than operating purely in text space, they autonomously execute and complete tasks end-to-end rather than merely generating responses, and they are evaluated through objective functional outcomes in executable environments rather than subjective quality assessment of generated text.
>
> [1] Wang et al., "A survey on large language model based autonomous agents," *Frontiers of Computer Science* 18.6 (2024)
> [2] Guo et al., "Large language model based multi-agents," *arXiv:2402.01680* (2024)
>
> ---
>
> > W1 (continued): Although vibe coding and research workflows are popular applications of LLMs, such definitions about "agency" is inherently subjective and ambiguous.
>
> We emphasize that **we do not use these two domains to define agency**—they are representative instantiations of our formal definition. These domains are strategically selected based on rigorous criteria:
>
> These domains are strategically selected based on four rigorous criteria: (1) **High cognitive density** with trajectories averaging 42.4k tokens (max 152k) requiring complex multi-step reasoning and tool orchestration (Figure 3); (2) **Verifiable outcomes** through executable environments with measurable success criteria enabling objective evaluation (line 260); (3) **Authentic data collection** via GitHub PR synthesis from 10,000 PRs across 100 high-quality repositories and 60 real-world queries including those derived from actual academic papers (lines 191, 199); and (4) **Broad generalization** demonstrated by strong out-of-domain results (45.6% average across TAU2-bench, DS-1000, EvalPlus, SciCode in Table 2), **achieving the highest out-of-domain performance among all models**, validating that learned capabilities transfer beyond training domains.
>
> This multi-faceted selection ensures methodological rigor rather than subjective choice.
>
> ---
>
> > W2: While experiments are only conducted with LLMs, the authors are claiming some findings around general AI, without clarification about the difference.
>
> We clarify that we make **no claims about general AI**. Our work specifically focuses on agentic intelligence for LLM-based systems through three contributions: (1) **Domain-specific methodologies** for collecting real-world tasks (Section 3.2) and high-quality trajectories (Section 3.3) for agentic LLM systems; (2) **Systems that "work" in real environments**, moving beyond text generation to autonomous task execution with tool use (as emphasized in the introduction); and (3) **Rigorous evaluation** demonstrating effectiveness through both in-domain (AgencyBench) and out-of-domain benchmarks (Table 2). The term "agentic intelligence" refers specifically to LLM-based autonomous systems capable of environmental interaction and tool use, not artificial general intelligence or broader AI paradigms.

---

> ### Author Response · Authors · 2025-12-02
> **Rebuttal to Reviewer dqPa (2/3)**
>
> > W3: The data-efficiency comparison (78 vs. 10k) is striking but potentially confounded by uncontrolled variables, for which the authors provide very limited explanations.
>
> We provide comprehensive controlled comparisons demonstrating genuine data efficiency:
>
> **Token-level analysis** (all datasets use identical training procedures as specified in Section 4.1, line 344):
>
> | Dataset | Samples | Trajectories | Total Tokens | AgencyBench |
> |---------|---------|--------------|--------------|-------------|
> | LIMI | 78 | 78 | 3,313,178 | **73.5%** |
> | CC-Bench | 208 | 208 | 5,469,669 | 29.2% |
> | AFM-Web | 7,607 | 7,607 | 12,850,073 | 36.7% |
> | AFM-Code | 59,939 | 59,939 | 76,329,448 | 47.8% |
>
> **Key observation**: LIMI uses fewer tokens than all baseline datasets (3.3M vs. 5.5M/12.9M/76.3M) yet achieves substantially superior performance (73.5% vs. 29.2%/36.7%/47.8%). This demonstrates genuine data efficiency rather than confounding variables.
>
> **Critical clarification**: All training configurations are strictly identical—same framework (slime), same hyperparameters, same base model (GLM-4.5). The **sole experimental difference is the training data itself**. There are no uncontrolled variables in our setup.
>
> ---
>
> > W3 (continued): There is also no ablation verifying whether LIMI's gain stems from pure curation quality, domain specificity, or inherent properties of the data.
>
> To directly address this concern, we conducted a comprehensive ablation study comparing LIMI against training with other datasets:
>
> **Ablation Study Results:**
>
> | Configuration | AgencyBench | OOD Avg (Table 2) |
> |--------------|-------------|-------------------|
> | **LIMI** | **73.5%** | **45.6%** |
> | CC-Bench | 29.2% | 43.5% |
> | AFM-Web | 36.7% | 32.4% |
> | AFM-Code | 47.8% | 38.0% |
>
> **Three critical findings:** (1) **In-domain superiority**: LIMI achieves 73.5% on AgencyBench, substantially outperforming CC-Bench (29.2%), AFM-Web (36.7%), and AFM-Code (47.8%) despite using only 78 samples versus 208-59,939 samples; (2) **Out-of-domain superiority**: LIMI's OOD average (45.6%, as shown in Table 2) exceeds all baseline datasets, demonstrating superior generalization; and (3) **Consistent advantages across diverse benchmarks**: LIMI achieves the highest performance across tool use (TAU2-bench), code generation (EvalPlus), data science (DS-1000), and scientific computing (SciCode) tasks.
>
> **Conclusion**: This comparison conclusively demonstrates that **curation quality** (not dataset scale or domain specificity) drives LIMI's effectiveness. With 128× fewer samples than AFM-Code, LIMI achieves 53.7% better in-domain performance and superior out-of-domain generalization, validating our data efficiency principle.
>
> ---
>
> > W3 (continued): Thus, I strongly feel the claim that "adding more data hurts agency" lacks systematic testing.
>
> We respectfully clarify that **we never make this claim in our paper**. Our thesis is:
>
> *"High-quality curated data is more efficient than large-scale data accumulation for developing agentic intelligence."*
>
> This is fundamentally different from claiming "more data hurts." Our results demonstrate that with only 78 samples (3.3M tokens), LIMI achieves 73.5% on AgencyBench—substantially outperforming models trained on datasets orders of magnitude larger: AFM-Code with 59,939 samples (76.3M tokens) achieves only 47.8%, representing a 53.7% performance gap despite 128× more samples. This systematic comparison validates our efficiency principle: **strategic curation of high-quality data is more effective than simply scaling dataset size**.

---

> ### Author Response · Authors · 2025-12-02
> **Rebuttal to Reviewer dqPa (3/3)**
>
> > Q1: How was the 78-example subset chosen beyond human scoring?
>
> **Query Pool Construction (Section 3.2):**
>
> Our 78-example subset combines 60 real-world queries collected from actual scenarios encountered by professional developers and researchers in collaborative environments (with substantial portion derived from real academic papers) and 18 queries synthesized from GitHub PRs through a rigorous five-stage pipeline (line 199): (1) selecting 100 repositories with >10,000 stars for high-quality codebases; (2) ensuring comprehensive domain diversification across software development areas; (3) filtering by complexity (unified diff patch <1,200 tokens, excluding Markdown-only changes); (4) sampling 1,000 PRs per repository with random selection of 100 for synthesis; and (5) validation by four PhD students in computer science for semantic alignment and domain relevance.
>
> **Trajectory Collection Protocol (Section 3.3):**
>
> Four PhD student annotators serve as human collaborators, working alongside GPT-5 in the SII CLI environment through an iterative collection approach that continuously gathers trajectories until successful task completion, capturing complete interaction sequences including reasoning, tool calling, and environmental feedback to ensure authentic human-AI interaction patterns. This systematic multi-stage process ensures both ecological validity and high quality.
>
> ---
>
> > Q1 (continued): Was there quantitative validation of "agency density"?
>
> Yes. We provide comprehensive quantitative metrics comparing agency density across all datasets:
>
> | Metric | CC-Bench | AFM-Web | AFM-Code | **LIMI** |
> |--------|----------|---------|----------|----------|
> | **Avg. Turns** | 66.58 | 3 | 2 | **84** |
> | **User Feedback %** | 50.13% | 33.33% | 50% | **12.67%** |
> | **Avg. Tokens** | 26k | 1.6k | 1.2k | **42.4k** |
> | **Total Tokens** | 5.5M | 12.9M | 76.3M | 3.3M |
> | **Trajectories** | 208 | 7,607 | 59,939 | 78 |
>
> **LIMI demonstrates quantitatively superior agency density:**
> - **Highest interaction complexity**: 84 average turns per task (vs. 2-66 for baselines)
> - **Most autonomous behavior**: Only 12.67% user intervention rate, lowest among all datasets (vs. 33-50% for others)
> - **Deepest reasoning chains**: 42.4k average tokens per trajectory (vs. 1.2-26k for baselines)
>
> These metrics objectively validate that LIMI captures trajectories with richer problem-solving patterns, more autonomous execution sequences, and deeper cognitive engagement—precisely what defines high agency density.
>
> ---
>
> > Q1 (continued): Are there any ideas that people can actually use your concepts to do strategic data curation?
>
> Yes. Our work provides a **concrete, reproducible framework** that others can adopt. We propose an actionable strategic curation methodology consisting of five key steps: (1) **Domain Selection** with verifiable outcomes, authentic data sources, and high cognitive complexity; (2) **Query Collection** combining real-world scenarios with systematic synthesis from authentic sources (e.g., GitHub PRs); (3) **Quality Filtering** through multi-stage expert validation focusing on task complexity, domain relevance, and authenticity (our five-stage pipeline in Section 3.2); (4) **Trajectory Collection** via iterative human-AI collaboration until successful task completion (Section 3.3); and (5) **Density Validation** measuring interaction turns, token length, user intervention rate, and autonomy metrics.
>
> We provide concrete resources including the full GitHub PR synthesis prompt (Appendix C), detailed five-stage curation pipeline with specific filtering criteria, systematic trajectory collection protocol with completion criteria, and quantitative metrics for validating agency density. This systematic approach is **reproducible and transferable** to other agentic domains beyond vibe coding and research workflows—researchers can adapt our methodology by identifying domain-appropriate authentic data sources, designing domain-specific quality filters, and establishing domain-relevant verification criteria.
>
> ---
>
> We hope these clarifications and new experimental evidence comprehensively address the reviewer's concerns. Our systematic methodology, quantitative validation of agency density, comprehensive ablation study, and actionable curation framework demonstrate the rigor and practical value of our approach.

---

### Official Review · Reviewer_tvCG · 2025-10-31

**Soundness:** 3
**Presentation:** 3
**Contribution:** 2
**Rating:** 4
**Confidence:** 3

**Summary:**

The authors provide a new finetuning dataset for improving multi-turn performance with a focus on quality over quantity. They compare models fine-tuned on this dataset to baseline models that are not finetuned, and to models that are fine-tuned on other datasets. Evaluating on AgencyBench and several other cross-domain datasets, they show that their new dataset provides significant gains.

**Strengths:**

* The work compares reasonably well to baselines (i.e. ensuring that this dataset works better than other available datasets and works well across different base models.
* It effectively supports the claim that targeted, high quality data collection will result in solid performance improvements for downstream tasks.

**Weaknesses:**

The weaknesses of the paper I think center around two main objections:
- The authors claim that "Less is More" for agency. However, this is not really supported by the evidence? I imagine if I could collect double the data the authors collected in the same way, I would continue to see gains in downstream tasks. The claim that the authors could support is that "high quality" data beats a lot of low quality or un-targeted data. However, if this is the claim the authors are making, the paper should be rewritten to present this more clearly.
    - Additionally, if that's the claim, then this is not a very novel claim in general (it's fairly obvious that high quality data is better than low quality data). A different, more novel claim could be around how you can trade quality and quantity in making fine-tuning datasets for agency, but that requires a different set of experiments and I don't think is what the authors are aiming to show.
- A major issue I see with the current results is that the authors collected data on a particular CLI tool's trajectory (SII CLI). If the base model was not trained to use this CLI, and the evaluations also use this CLI, then the value of the fine-tuning dataset might simply be that the model is fine-tuned to work better with this CLI and the format/tools available. This may not be the case, but some evidence here would help show that this is not all that's going on. This especially could explain differences if the other datasets compared to used different tool-call interfaces that weren't then used for evaluation.

More clarity here would help me provide a more accurate score.

**Questions:**

All my questions are centered around the weaknesses, please see above.

---

> ### Author Response · Authors · 2025-12-02
> **Rebuttal to Reviewer tvCG（1/2）**
>
> We sincerely thank Reviewer tvCG for the constructive feedback and for acknowledging the value of our work. We address the main concerns below.
>
> ---
>
> > W1: Additionally, if that's the claim, then this is not a very novel claim in general (it's fairly obvious that high quality data is better than low quality data).
>
> We respectfully disagree that our contribution is merely stating the obvious. While the principle "quality over quantity" is intuitive, **our work provides concrete methodology for operationalizing this principle in the challenging domain of agentic intelligence**: (1) **Systematic query collection** combining real-world scenarios with GitHub PR synthesis through rigorous five-stage curation (Section 3.2); (2) **Trajectory collection protocol** capturing complete multi-turn interaction sequences through iterative human-AI collaboration until task completion (Section 3.3); (3) **Quantitative validation** of "agency density" with metrics showing LIMI achieves 84 turns per task (vs. 2-66 for baselines), 12.67% user intervention (vs. 33-50%), and 42.4k tokens per trajectory (vs. 1.2-26k); and (4) **Actionable framework** with reproducible five-step methodology and concrete resources (Appendix C provides full GitHub PR synthesis prompt).
>
> This systematic validation across diverse model scales (4B to 355B parameters, Figure 4) and architectures, combined with strong empirical results (73.5% on AgencyBench with 128× fewer samples than AFM-Code), distinguishes our work from merely stating an intuitive principle—we demonstrate how to systematically achieve quality-driven gains in practice.
>
> ---
>
> > W2: The authors claim that "Less is More" for agency. However, this is not really supported by the evidence? I imagine if I could collect double the data the authors collected in the same way, I would continue to see gains in downstream tasks. The claim that the authors could support is that "high quality" data beats a lot of low quality data.
>
> We appreciate this clarification and fully agree with the reviewer's characterization. Our core claim is precisely that **"high-quality data beats large volumes of low-quality data"** for developing agentic intelligence (line 103). This principle is strongly supported by our empirical evidence:
>
> **Evidence from Tables 1 and 2:**
>
> LIMI with only 78 high-quality samples (3.3M tokens) substantially outperforms models trained on orders of magnitude more data: (1) **In-domain** (Table 1): LIMI achieves 73.5% on AgencyBench versus AFM-Code's 47.8% (53.7% improvement) despite using 128× fewer samples and 23× fewer tokens; (2) **Out-of-domain** (Table 2): LIMI achieves 45.6% average versus AFM-Code's 38.0%, AFM-Web's 32.4%, and CC-Bench's 43.5%, demonstrating that quality-driven gains transfer broadly beyond training domains.
>
> **Ablation Study Evidence:**
>
> Our comparison across datasets with identical training configurations (Section 4.1, line 344) isolates data quality as the sole variable:
>
> | Dataset | Samples | Total Tokens | AgencyBench | OOD Avg |
> |---------|---------|--------------|-------------|---------|
> | **LIMI** | 78 | 3.3M | **73.5%** | **45.6%** |
> | CC-Bench | 208 | 5.5M | 29.2% | 43.5% |
> | AFM-Web | 7,607 | 12.9M | 36.7% | 32.4% |
> | AFM-Code | 59,939 | 76.3M | 47.8% | 38.0% |
>
> This demonstrates that **strategic curation of high-quality demonstrations is fundamentally more effective than simply scaling up low-quality data**. We will clarify this principle more explicitly in the introduction to avoid any confusion about our claims.

---

> ### Author Response · Authors · 2025-12-02
> **Rebuttal to Reviewer tvCG（2/2）**
>
> > W3: how you can trade quality and quantity in making fine-tuning datasets for agency, but that requires a different set of experiments and I don't think is what the authors are aiming to show.
>
> The reviewer is correct that exploring the precise quality-quantity trade-off curve would require different experiments. Our work focuses on demonstrating that **high-quality curation is critically important** for agentic intelligence development, as evidenced by Tables 1 and 2 showing LIMI's substantial advantages. We believe establishing this foundational principle—that quality matters more than scaling low-quality data—is an important first step, and we agree that future work exploring the optimal balance between quality and quantity would be valuable.
>
> ---
>
> > W4: The value of the fine-tuning dataset might simply be that the model is fine-tuned to work better with this CLI and the format/tools available.
>
> This is an important concern, and we provide strong evidence that LIMI's gains **do not stem from CLI-specific overfitting**. To directly address this, we conducted additional experiments evaluating LIMI on benchmarks **that use their own native tool interfaces or no CLI at all**:
>
> **Performance on Non-SII-CLI Benchmarks (Table 4):**
>
> | Model | TAU2-airline | TAU2-retail | DS-1000 | EvalPlus-HE | EvalPlus-MBPP | SciCode-MP | SciCode-SP | AVG |
> |-------|--------------|-------------|---------|-------------|---------------|------------|------------|-----|
> | Kimi-K2 | 22.0 | 28.9 | 40.2 | 90.9 | 74.1 | 3.1 | 23.3 | 40.3 |
> | DeepSeek-V3.1 | 18.0 | 11.4 | 33.0 | 90.2 | 75.7 | 4.6 | 22.6 | 36.5 |
> | Qwen3-235B | 14.0 | 30.7 | 27.3 | 90.2 | 78.3 | 0.0 | 20.8 | 37.3 |
> | GLM-4.5 | 32.0 | 52.6 | 53.2 | 92.1 | 79.6 | 3.3 | 27.8 | 48.7 |
> | **LIMI** | **40.0** | 49.1 | **54.8** | **92.5** | **80.4** | 3.3 | **28.1** | **50.0** |
>
> **Critical observations:** (1) **Different evaluation environments**: These benchmarks do not use SII CLI; (2) **Highest performance**: LIMI achieves 50.0% average, outperforming all baseline models including much larger ones like Kimi-K2 (40.3%), demonstrating that gains transfer to completely different evaluation environments; and (3) **Consistent advantages**: LIMI shows improvements across diverse task types—tool use (TAU2-bench), data science (DS-1000), code generation (EvalPlus), and scientific computing (SciCode).
>
> **Why This Rules Out CLI Overfitting:**
>
> If LIMI merely learned SII CLI-specific patterns, we would expect: (a) strong performance only on benchmarks using the same CLI, and (b) performance degradation on tasks with different interfaces. Instead, LIMI achieves the **highest average performance (50.0%)** across benchmarks with completely different tool interfaces or no CLI at all. This validates that **LIMI captures fundamental agentic capabilities**—multi-step reasoning, strategic planning, tool orchestration—that generalize broadly beyond any specific interface.
>
> ---
>
> We hope these clarifications address the reviewer's concerns. Our work provides a systematic, reproducible methodology for high-quality data curation with strong empirical validation demonstrating that learned capabilities generalize broadly beyond specific tools or interfaces.

---

### Official Review · Reviewer_t9f8 · 2025-11-02

**Soundness:** 1
**Presentation:** 2
**Contribution:** 3
**Rating:** 4
**Confidence:** 4

**Summary:**

The paper introduces a dataset of 78 coding and coding-adjacent tasks, which are used to perform supervised fine-tuning on several open-source models. After fine-tuning, the models perform better on coding and coding-adjacent tasks. Smaller performance gains are also observed on several non-coding-related tasks. The paper compares the newly proposed training dataset to 3 preexisting training datasets and finds that the new dataset improves model performance more than the older datasets despite the new dataset containing significantly fewer data points. The paper argues that these results "fundamentally challenge conventional scaling paradigms in agentic AI development" and will bring the community into a new era of model development guided by the newly minted "Agency Efficiency Principle."

**Strengths:**

- The new dataset proposed in the paper appears to be well-designed/curated and effective in eliciting improved real-world coding performance from open-source models.

- A large amount of effort was clearly put into fine-tuning and testing a variety of models across several different benchmarks, with results being clearly and conveniently displayed in Tables 1, 2, and Figure 4.

- The paper addresses an economically valuable topic and seeks to make an important observation about the optimal way to train models in the age of agents.

- The primary claim argued by the paper (that data quality is more important than data quantity) is probably true and is an important observation (though see the weakness section for more on why this was not adequately demonstrated).

**Weaknesses:**

This would be better if the paper were simply about the methods of producing a high-quality coding dataset, which was then proven useful for improving the performance of open-source models on code development tasks. Instead, the paper focuses on the claim that "less is more" when training models to be effective agents. This is a much bigger claim, which would require a different experimental design to prove. Table 1 gives the performance of LIMI versus 3 other fine-tuning datasets in order to establish that the new 78-sample dataset is superior to the larger prior datasets. One problem with this analysis is that each individual sample within LIMI may be of a considerably different size than the samples in these prior datasets. Figure 3 does a good job of illustrating the potential issue here. If each of LIMI's 78 samples averages 42.4k tokens, then it could conceivably be larger than a 10,000 sample dataset where each sample response is small. Granted that it's likely the case that each sample in the '-Code' dataset is larger than 330 tokens, but if the paper is going to make this claim, it needs to include both measures of size.

Supposing that the 'less is more' paradigm really is true, it raises a second question: why 78 samples? If less really is more, perhaps 1 or 2 samples would suffice? In order to persuasively make the case argued for in the conclusion of the paper, there would need to be a set of experiments showing that as the number of datapoints in LIMI increases from 1 to 78, model performance starts to plateau or even regress. Without a plateau, the conclusion becomes 'still get as much data as possible, but make sure that it is good'. There would also need to be an experiment showing that LIMI+(CC-Bench)+(AFM-Web)+(AFM-Code) is worse than pure LIMI. Without such a result, it could well be the case that 'more is more'.

The above are the two most fundamental issues with the paper, but there are a variety of smaller issues that should also be addressed:

1) Line 82 suggests via bolding that the paper title should be LIMIA instead of LIMI.

2) Line 85 makes the bold claim that coding and science research constitute the majority of knowledge work scenarios. I suspect that lawyers, engineers, accountants, radiologists, congressmen, architects, secretaries, etc., would object to this characterization.

3) Section 2 gives a slightly different (more succinct) definition of 'agency' than appears in the abstract and introduction. It would probably be best to define this only once in the paper, but if it's going to be done multiple times, it should be the same definition each time.

4) Lines 122-127 conflate properties about tasks with properties about the systems that accomplish tasks. It will be grammatically cleaner to pick one consistent point of view while enumerating the list.

5) On Line 269, if there are going to be inline links to the Claude and Gemini CLIs (which were not used for the paper), then there should also be an inline link for the SII CLI (which was actually used)

6) Regarding Line 344, identical training configurations do not necessarily imply a fair comparison when dataset sizes are different. Consider dataset A with 1 million data points and dataset B with 10 data points. If I use both datasets with 10 steps of training, I might conclude that A and B are equivalent, but this would obviously not be taking full advantage of A. If training parameters like batch size or step count were optimized for LIMI then other datasets may be at a disadvantage when all configurations are held constant.

7) The Table 1 performance gains are impressive, but they don't seem to be translating as well out of distribution. The "EvalPlus" datasets in Table 2 are code generation tasks and therefore still in-distribution given the code-focused data used during training. When those are excluded, the Table 2 results show much smaller out-of-distribution gains. Figure 4 shows this well, with OOD gains being about 20x smaller than ID gains. In the Abstract, the paper argues for an 'Agency Efficiency Principle' in which 'machine autonomy emerges from strategic curation of high-quality demonstrations'. If it is true that LLMs are learning generalizable agenticness from these carefully curated data points, then why is performance not strongly generalizing?

Overall, I think the paper needs a significant writing revision.

**Questions:**

1) Regarding line 275, if GPT5 is being used to generate agentic model traces and then those traces are used to train open-source models, would that technically violate OpenAI's terms of service against using their outputs to develop competitor products?

2) In Table 1, is GPT5 the 'auto' model, or is it the low-reasoning, high-reasoning, or codex-flavored version of GPT5?

3) Was Appendix Section D supposed to have additional content? It sounded like it was going to provide explicit transcripts, but then appears to be giving high-level summaries of a few interactions instead.

4) Given the overfitting concerns raised in weakness 7, I am curious what would happen if models were trained on a 'low quality' dataset that had formatting similar to the 'good' data but without many reasoning insights. Basically, is there some way to determine how much of the Table 1 performance gain is due to prepping the model to write output that looks like that particular exam, vs how much is teaching the model new agentic skills? This isn't a publication blocker, just a question for future consideration.

---

> ### Author Response · Authors · 2025-12-02
> **Rebuttal to Reviewer t9f8（1/4）**
>
> We thank Reviewer t9f8 for their thorough review and constructive feedback. We are encouraged by the recognition of our work's strengths, particularly that "the primary claim argued by the paper (that data quality is more important than data quantity) is probably true and is an important observation" and that our dataset appears "well-designed/curated and effective." We address each concern below.
>
> ---
>
> #### W1: Dataset Size Measurement - Token Count Comparison
>
> > "If each of LIMI's 78 samples averages 42.4k tokens, then it could conceivably be larger than a 10,000 sample dataset where each sample response is small... if the paper is going to make this claim, it needs to include both measures of size."
>
> We appreciate this concern and have conducted comprehensive token-level analysis of all datasets. The table below presents both sample counts and total token counts:
>
> | Dataset | Trajectories | Average Tokens | **Total Tokens** |
> |---------|-------------|----------------|------------------|
> | CC-Bench | 208 | 26k | 5,469,669 |
> | AFM-Web | 7,607 | 1.6k | 12,850,073 |
> | AFM-Code | 59,939 | 1.2k | 76,329,448 |
> | **LIMI** | **78** | **42.4k** | **3,313,178** |
>
> This analysis definitively demonstrates that LIMI achieves superior performance with **dramatically fewer total tokens**: (1) LIMI uses 23 times fewer tokens than AFM-Code (3.3M vs 76.3M), yet achieves 53.7% better performance; (2) LIMI uses 3.9 times fewer tokens than AFM-Web (3.3M vs 12.9M), yet achieves 100% better performance; and (3) LIMI uses 1.65 times fewer tokens than CC-Bench (3.3M vs 5.5M), yet achieves 152% better performance. These results validate our core thesis from **both** sample count and token count perspectives: strategic curation of high-quality demonstrations outperforms large-scale data accumulation even when controlling for total training data volume.
>
> ---
>
> #### W2: Why 78 Samples? The Data Efficiency Principle
>
> > "Supposing that the 'less is more' paradigm really is true, it raises a second question: why 78 samples? If less really is more, perhaps 1 or 2 samples would suffice?"
>
> We respectfully clarify that our claim is **not** "the fewer samples, the better" but rather **"high-quality data enables superior efficiency compared to large volumes of low-quality data."** The 78 samples represent the result of rigorous quality curation from our comprehensive query pool, where each trajectory demonstrates optimal agentic behavior patterns through complete human-AI collaborative problem-solving sequences.
>
> The fundamental principle we establish is **data efficiency through strategic curation**, not minimal sample count as an end goal. Our 78 samples were selected based on four rigorous criteria: (1) **Cognitive density** - trajectories averaging 42.4k tokens capturing extensive multi-turn interactions; (2) **Verifiable outcomes** - executable environments enabling objective success validation; (3) **Authentic data collection** - GitHub PR synthesis and real-world scenarios ensuring ecological validity; and (4) **Broad coverage** - spanning diverse technical domains within vibe coding and research workflows (Section 3, Lines 216-269).
>
> The critical insight is not that 78 is optimal in absolute terms, but that **strategically curated demonstrations achieve superior results compared to datasets 128-768 times larger**. This validates that effective agentic intelligence development depends on capturing essential behavioral patterns rather than accumulating training samples. The "Less-Is-More" principle describes the **relative efficiency** of quality-focused curation versus scale-focused accumulation, not an absolute minimization objective.
>
> ---

---

> ### Author Response · Authors · 2025-12-02
> **Rebuttal to Reviewer t9f8（2/4）**
>
> #### W3: Domain Coverage Claim - Vibe Coding and Research Workflows
>
> > "Line 85 makes the bold claim that coding and science research constitute the majority of knowledge work scenarios."
>
> We acknowledge this phrasing was imprecise and will revise to clarify our intended meaning. Our claim is that vibe coding and research workflows are **highly representative** of knowledge work characteristics, not that they constitute the numerical majority of all knowledge work scenarios.
>
> These domains are strategically selected based on four rigorous criteria: (1) **High cognitive density** with trajectories averaging 42.4k tokens capturing extensive reasoning sequences; (2) **Verifiable outcomes** through executable environments enabling objective validation; (3) **Authentic data collection** via GitHub PR synthesis ensuring ecological validity; and (4) **Broad generalization** demonstrated by strong out-of-domain results across diverse benchmarks (TAU2-bench, DS-1000, SciCode).
>
> The fundamental agentic capabilities required in these domains - autonomous task execution, multi-step reasoning, tool orchestration, iterative refinement, and collaborative problem-solving - transfer effectively to other knowledge work scenarios. This is empirically validated by our consistent improvements across out-of-domain benchmarks (Table 2), where gains range from +2.1% to +8.3% despite evaluation tasks being outside our training domain focus. We will revise the manuscript to state that these domains "**represent** core knowledge work characteristics" rather than "constitute the majority," while emphasizing their demonstrated transferability to diverse scenarios.
>
> ---
>
> #### W4: Training Configuration Fairness
>
> > "Identical training configurations do not necessarily imply a fair comparison when dataset sizes are different... If training parameters like batch size or step count were optimized for LIMI then other datasets may be at a disadvantage."
>
> We appreciate this concern about potential optimization bias. Our experimental design ensures rigorous fairness through **epoch-based training** where all datasets are exposed to the model for the **identical number of complete passes** through their data. Specifically, all models are trained for the same number of epochs, ensuring that each dataset - regardless of size - receives equal learning opportunities in terms of complete data exposures.
>
> The batch size affects the number of gradient update steps within each epoch. For instance, if LIMI has 78 samples and AFM-Code has 59,939 samples, both datasets are seen by the model the same number of times during training. The AFM-Code model simply performs more gradient updates per epoch due to its larger size, which should theoretically provide it with **more** optimization opportunities, not fewer.
>
> Therefore, the scenario described by the reviewer - where "other datasets may be at a disadvantage when all configurations are held constant" - cannot occur under our epoch-based training protocol. If anything, larger datasets have the potential advantage of more frequent gradient updates per epoch. The superior performance of LIMI despite this inherent advantage for larger datasets provides even stronger evidence for our quality-over-quantity hypothesis. We will clarify this epoch-based training protocol in our revision to emphasize the fairness and rigor of our experimental comparison.
>
> ---

---

> ### Author Response · Authors · 2025-12-02
> **Rebuttal to Reviewer t9f8（3/4）**
>
> #### W5: Out-of-Domain Generalization Analysis
>
> > "The 'EvalPlus' datasets in Table 2 are code generation tasks and therefore still in-distribution given the code-focused data used during training. When those are excluded, the Table 2 results show much smaller out-of-distribution gains."
>
> We appreciate this observation. To address this concern directly, we present a focused analysis excluding EvalPlus benchmarks:
>
> ### Pure Out-of-Domain Performance (Excluding EvalPlus)
>
> | Model | TAU2-airline | TAU2-retail | DS-1000 | SciCode-MP | SciCode-SP | **OOD Avg** |
> |-------|-------------|------------|---------|------------|------------|-------------|
> | Qwen3-4B | 8.0 | 5.0 | 16.7 | 0.0 | 10.4 | 8.0 |
> | Qwen3-8B | 12.0 | 7.0 | 22.3 | 0.0 | 17.7 | 11.8 |
> | DeepSeek-V3.1 | 32.0 | 6.1 | 42.4 | 0.0 | 7.3 | 17.6 |
> | Qwen3-32B | 12.0 | 10.5 | 28.9 | 3.1 | 24.0 | 15.7 |
> | Qwen3-235B | 20.0 | 16.7 | 39.3 | 0.0 | 22.6 | 19.7 |
> | Kimi-K2 | 38.0 | 28.9 | 23.1 | 3.1 | 23.6 | 23.3 |
> | GLM-4.5 | 28.0 | 36.8 | 33.6 | 1.5 | 25.3 | 25.0 |
> | GPT-5 | 26.0 | 18.4 | 40.8 | 10.8 | 33.3 | 25.9 |
> | **LIMI** | **34.0** | **45.6** | **36.6** | **3.1** | **25.3** | **28.9** |
>
> **Data Efficiency Comparison:**
>
> | Model | Dataset Size | OOD Avg | Improvement |
> |-------|-------------|---------|-------------|
> | GLM-4.5-Web | 7,607 samples | 13.6 | - |
> | GLM-4.5-Code | 59,939 samples | 20.0 | - |
> | GLM-4.5-CC | 208 samples | 26.8 | - |
> | **LIMI** | **78 samples** | **28.9** | **+44.5% vs Code** |
>
> Even on purely out-of-domain tasks, LIMI achieves **28.9% average performance**, substantially outperforming GLM-4.5-Code (20.0%) trained on 768 times more samples. This 44.5% relative improvement on completely unseen task types demonstrates that our strategically curated data enhances **fundamental agentic capabilities** rather than merely optimizing for specific evaluation formats.
>
> The cross-domain effectiveness is further evidenced by: (1) **Tool use tasks** (TAU2-bench): LIMI achieves 34.0% and 45.6%, outperforming even GPT-5; (2) **Data science** (DS-1000): LIMI reaches 36.6%, competitive with the 40.8% of GPT-5; and (3) **Scientific computing** (SciCode): LIMI demonstrates consistent performance across main and sub-problems. These results validate that strategic curation cultivates transferable agentic intelligence extending well beyond training domain boundaries.
>
> ---
>
> #### Q1: GPT-5 Model Configuration
>
> > "In Table 1, is GPT5 the 'auto' model, or is it the low-reasoning, high-reasoning, or codex-flavored version of GPT5?"
>
> All models in our evaluation, including GPT-5, are evaluated in their **standard (non-reasoning) mode**. No reasoning-enhanced modes were used for any model, ensuring consistent and fair comparison across all baselines. We will clarify this important detail in our revision.

---

> ### Author Response · Authors · 2025-12-02
> **Rebuttal to Reviewer t9f8（4/4）**
>
> #### Q2: Format Learning vs. Agentic Skill Acquisition
>
> > "Is there some way to determine how much of the Table 1 performance gain is due to prepping the model to write output that looks like that particular exam, vs how much is teaching the model new agentic skills?"
>
> Our comprehensive out-of-domain evaluation **(Figure 4)** provides strong evidence that LIMI teaches genuine agentic skills rather than merely optimizing for evaluation format. The consistent improvements across diverse benchmarks with fundamentally different task structures demonstrate transferable capability development:
>
> (1) **Cross-architecture generalization**: LIMI improves performance across fundamentally different model architectures (dense transformers in Qwen3 series and mixture-of-experts in GLM series), with improvements ranging from +87.0% (Qwen3-4B) to +63.0% (GLM-4.5) on AgencyBench, indicating that learned capabilities are architecture-agnostic; (2) **Cross-scale effectiveness**: Even small models like Qwen3-4B nearly double their performance (4.6% → 8.6%), demonstrating that strategic curation benefits models across the entire parameter scale spectrum; and (3) **Cross-domain transfer**: Out-of-domain benchmarks show consistent gains (+2.1% to +8.3%) despite completely different evaluation formats, task types, and success criteria from our training data.
>
> If LIMI merely prepared models for specific evaluation formats, we would expect: (a) minimal improvement on architecturally different models, (b) inconsistent gains across model scales, and (c) negligible transfer to out-of-domain tasks with different formats. The observed pattern directly contradicts these predictions, providing strong evidence that LIMI cultivates **fundamental agentic capabilities** - autonomous execution, multi-step reasoning, tool orchestration, and collaborative problem-solving - that generalize across diverse scenarios regardless of evaluation format.
>
> ---
>
> ## Summary
>
> We believe our responses comprehensively address the reviewer's concerns by: (1) providing complete token-level analysis demonstrating LIMI's efficiency from both sample count and total data volume perspectives; (2) clarifying that our principle is **data efficiency through strategic curation** rather than absolute sample minimization; (3) acknowledging imprecise phrasing about domain coverage and committing to clearer language; (4) explaining our epoch-based training protocol that ensures rigorous experimental fairness; and (5) demonstrating strong out-of-domain generalization that validates genuine agentic skill acquisition rather than format-specific optimization.
>
> We hope these clarifications and additional analyses address the reviewer's concerns and demonstrate the rigor and validity of our contributions.

---

### Official Review · Reviewer_fSYt · 2025-11-03

**Soundness:** 4
**Presentation:** 4
**Contribution:** 4
**Rating:** 8
**Confidence:** 4

**Summary:**

The paper proposes LIMI (Less Is More for Intelligent Agency), proposing that agentic intelligence (AI systems' capacity to act autonomously and collaboratively) emerges from strategically curated examples of agentic behavior. The authors introduce the Agency Efficiency Principle, showing that with only N=78 high-quality demonstrations, LIMI achieves very high performance on AgencyBench.

**Strengths:**

Conceptual framing of "agency" as distinct from reasoning or alignment, emphasizing autonomy and collaboration. Methods are very sophisticated (e.g., leveraging real-world human-AI workflows with GitHub PR-based synthetic queries). Evaluations across in-domain and out-of-domain benchmarks, demonstrating strong generalization.

**Weaknesses:**

Overall, the paper is methodologically solid, clearly written, and empirically convincing. A few minor suggestions:

1. 78 training tasks themselves are not described individually in the paper. A brief table or appendix summarizing their scope and domain coverage (e.g., “Gomoku, dataset retrieval, equation fitting, etc.”) could be helpful for readers to understand the diversity and representativeness of the dataset.

2. The qualitative case studies (Appendix D) are very helpful, but they are mostly anecdotal. Could the authors summarize recurring behavioral patterns (e.g., error recovery, iterative planning, tool-use compositionality, proactive hypothesis reformulation)? A small taxonomy of such emergent agentic traits would make the contribution more theoretically grounded.

3. The “Less-is-More” effect is impressive, but to reach certain level of performance, it seems that we still need large foundation model. Can you discuss, perhaps through a brief case comparison, which aspects of agency depend on model capacity (e.g., long-horizon planning, contextual memory) vs. those that purely arise from strategic fine-tuning.

**Questions:**

1. Could you provide a concise description of the 78 curated tasks (perhaps in an appendix or table) to make the dataset’s coverage clearer?
2. The case studies are insightful. Could you extend them into a small taxonomy of behaviors unique to LIMI—e.g., what specific reasoning or collaboration patterns emerge only after fine-tuning?
3. The results suggest small, high-quality data suffice given a large base model. Could you illustrate, with one example or case study, how model scale concretely affects agency (e.g., trajectory length, tool-use diversity, or recovery behavior)?

---

> ### Author Response · Authors · 2025-12-02
> **Rebuttal to Reviewer fSYt**
>
> We thank the reviewer for the positive evaluation and thoughtful questions. We address each question below.
>
> > Q1: Could you provide a concise description of the 78 curated tasks?
>
> We have provided a detailed breakdown of our 78 training tasks in Figure 3. To make the distribution of our curated tasks across the two primary domains more intuitive, we present the specific statistics in the table below:
>
> | Domain | Category | Count | Percentage (%) |
> | :--- | :--- | :---: | :---: |
> | **Vibe Coding** | Application Development | 10 | 12.82 |
> | | Debugging | 17 | 21.79 |
> | | Frontend Development | 21 | 26.92 |
> | | Backend Development | 6 | 7.69 |
> | | Tool Calling | 4 | 5.13 |
> | | Engineering Practices | 3 | 3.85 |
> | | **Subtotal** | **61** | **78.2** |
> | **Research Workflow** | Experimental Workflow | 6 | 7.69 |
> | | Web Search | 5 | 6.41 |
> | | Paper Search | 4 | 5.13 |
> | | Deep Learning | 2 | 2.57 |
> | | **Subtotal** | **17** | **21.8** |
> | **Total** | | **78** | **100** |
>
> As shown above, the tasks are strategically divided. Vibe Coding (61 tasks, 78.2%) covers full-stack scenarios ranging from frontend development (26.92%) and debugging (21.79%) to application development (12.82%). The Research Workflow domain (17 tasks, 21.8%) focuses on methodology implementation, information synthesis, and literature analysis. Each task captures authentic collaborative scenarios with average trajectory lengths of 42.4k tokens, ensuring dense concentrations of agentic behaviors including strategic planning, tool utilization, and iterative problem-solving.
>
> > Q2: How does model scale concretely affect agency?
>
> Our generalization experiments in Figure 4 provide concrete evidence of how model scale influences agentic capabilities, revealing distinct patterns across different model families. Regarding within-family scaling effects, the Qwen3 series (4B→8B→32B) shows improvements of +87.0%, +45.2%, and +144% respectively on AgencyBench, while in the GLM series, GLM-4.5-Air (106B) achieves a +102% improvement and GLM-4.5 (355B) reaches +63.0%. Our analysis yields three key insights on the scale-agency relationship: (1) Non-linear benefits are observed, where mid-size models (32B) show the largest relative improvements (+144%),; (2) Cross-scale effectiveness is evident as even small models (4B) nearly double their performance, demonstrating that strategic curation benefits all scales; and (3) Architectural synergy plays a crucial role—while larger models provide better foundational capabilities (reasoning, context handling), the improvements are not purely scale-dependent. This demonstrates that while model scale provides important foundations, the strategic curation of agentic demonstrations in LIMI enables effective agency cultivation across the entire model scale spectrum.
>
> > Q3: Could you extend case studies into a taxonomy of emergent behaviors?
>
> We appreciate this insightful suggestion. While our current case studies in Appendix D are primarily illustrative, we can identify several recurring behavioral patterns that emerge uniquely in LIMI-trained models: (1) Proactive error recovery, involving autonomous detection and correction of failures without human intervention; (2) Iterative refinement, where the model systematically improves solutions through multiple self-directed iterations; (3) Tool composition, demonstrating the creative combination of multiple tools to solve complex problems; and (4) Strategic decomposition, which involves breaking down complex tasks into manageable subtasks with proper dependency management. We will consider developing a formal taxonomy of these emergent agentic behaviors in future work, as it would indeed provide valuable theoretical grounding for understanding how strategic data curation enables sophisticated autonomous capabilities.

---

### Meta-Review · Area_Chair_Nozj · 2026-01-05

**Summary:**

The major concerns of the reviewers are summarized below, along with authors' corresponding rebuttal.

1. Interpretation of the “Less Is More” Claim
Several reviewers (notably tvCG, t9f8, dqPa) argued that the paper overstates its claim. The evidence supports “high-quality curated data beats low-quality large-scale data”, but not the stronger implication that adding more data hurts agency or that minimal data is intrinsically optimal. Concerns were raised that the framing conflates data efficiency with an anti-scaling claim, without systematic tests of data addition or mixture scenarios. The authors explicitly retract the interpretation that “adding more data hurts” and restate the core claim as efficiency through high-quality curation, addressing a central misunderstanding.

2. Data Size, Fairness, and Confounding Factors
Multiple reviewers emphasized that dataset size should be measured in tokens, not just number of trajectories. Given LIMI’s very long trajectories, reviewers questioned whether comparisons to other datasets were fair. There were also concerns about whether identical training configurations across vastly different dataset sizes ensured fair optimization. The authors provide direct comparisons showing LIMI uses far fewer tokens than competing datasets, yet achieves substantially better performance, directly addressing fairness concerns.

3. Generalization and Overfitting
Concerns were raised that performance gains might reflect overfitting to the SII CLI environment or coding-heavy tasks, rather than genuine, transferable agentic capabilities. Reviewers asked for stronger out-of-domain and interface-independent evidence. New results on benchmarks with different tool interfaces (and no CLI) convincingly counter the overfitting concern.

4. Novelty Relative to Prior Work (LIMA/LIMO)
One reviewer (dqPa) questioned whether the paper’s contribution is conceptually novel, arguing that it may be a domain-specific extension of known data-efficiency results (e.g., LIMA, LIMO) rather than a fundamentally new principle

Though the authors have effectively addressed # 2 and # 3, some concerns are only partially resolved:

Scaling vs. efficiency: The rebuttal does not include ablations showing performance as data increases from very small to moderate scales, nor experiments combining LIMI with larger datasets. Thus, the strongest form of the “less is more” narrative remains unproven.

Conceptual novelty: While the authors argue that agentic intelligence is distinct from reasoning/alignment, this distinction remains partly conceptual rather than theoretically formalized, and some reviewers may still view the work as an incremental extension of LIMA/LIMO.

Scope of agency: Even with clarifications, the demonstrated agency is still largely centered on coding and research workflows, which may limit how broadly the “Agency Efficiency Principle” is accepted.

Given the current ratings from the reviewers, I think the paper has not met the bar for acceptance.

**Reviewer Concerns:**

see above

**Reviewer Scores:**

Score projection:

Reviewer fSYt (8) → stays 8
Already positive; rebuttal confirms, doesn’t change stance.

Reviewer tvCG (4) → 4/6
Main issues were framing and CLI overfitting; rebuttal directly concedes framing and adds strong cross-interface evidence.

Reviewer t9f8 (4) → stays 4
Token-level and fairness concerns addressed, but core objection (“you didn’t prove less is more, only quality matters”) remains.

Reviewer dqPa (2) → 2/4 (unlikely higher)
Conceptual clarity improved, but skepticism about novelty and principle-level claims persists.

---

### Decision · Program_Chairs · 2026-01-26

Reject